# DiffPack: A Torsional Diffusion Model for Autoregressive Protein Side-Chain Packing

**Yangtian Zhang**[1,2 *]    **Zuobai Zhang**[1,2 *]    **Bozitao Zhong**[1,2]
**Sanchit Misra**[3]    **Jian Tang**[1,4,5 †]

[*]equal contribution    [†]corresponding author
[1]Mila - Québec AI Institute    [2]Université de Montréal    [3]Intel
[4]HEC Montréal    [5]CIFAR AI Research Chair
**contacts:** <yangtian.zhang, zuobai.zhang>@mila.quebec,  jian.tang@hec.ca

## Abstract

Proteins play a critical role in carrying out biological functions, and their 3D structures are essential in determining their functions. Accurately predicting the conformation of protein side-chains given their backbones is important for applications in protein structure prediction, design and protein-protein interactions. Traditional methods are computationally intensive and have limited accuracy, while existing machine learning methods treat the problem as a regression task and overlook the restrictions imposed by the constant covalent bond lengths and angles. In this work, we present **DiffPack**, a torsional diffusion model that learns the joint distribution of side-chain torsional angles, the only degrees of freedom in side-chain packing, by diffusing and denoising on the torsional space. To avoid issues arising from simultaneous perturbation of all four torsional angles, we propose autoregressively generating the four torsional angles from $\chi_1$ to $\chi_4$ and training diffusion models for each torsional angle. We evaluate the method on several benchmarks for protein side-chain packing and show that our method achieves improvements of 11.9% and 13.5% in angle accuracy on CASP13 and CASP14, respectively, with a significantly smaller model size ($60\times$ fewer parameters). Additionally, we show the effectiveness of our method in enhancing side-chain predictions in the AlphaFold2 model. Code is available at https://github.com/DeepGraphLearning/DiffPack.

## 1 Introduction

Proteins are crucial for performing a diverse range of biological functions, such as catalysis, signaling, and structural support. Their three-dimensional structures, determined by amino acid arrangement, are crucial for their function. Specifically, amino acid side-chains play a critical role in the stability and specificity of protein structures by forming hydrogen bonds, hydrophobic interactions, and other non-covalent interactions with other side-chains or the protein backbone. Therefore, accurately predicting protein side-chain conformation is an essential problem in protein structure prediction [16, 15, 11], design [48, 54, 14, 65] and protein-protein interactions [64, 24].

Despite recent advancements in deep learning models inspired by AlphaFold2 for predicting the positions of protein backbone atoms [32, 4], predicting the conformation of protein side-chains remains a challenging problem due to the complex interactions between side chains. In this work, we focus on the problem of predicting side-chain conformation with fixed backbone structure, *a.k.a.*, protein side-chain packing. Traditional methods for side-chain prediction rely on techniques such as rotamer libraries, energy functions, and Monte Carlo sampling  [27, 73, 37, 3, 1, 35, 71, 7]. However, these methods are computationally intensive and struggle to accurately capture the complex energy landscape of protein side chains due to their reliance on search heuristics and discrete sampling.

37th Conference on Neural Information Processing Systems (NeurIPS 2023).

Several machine learning methods have been proposed for side-chain prediction, including DL-Packer [46], AttnPacker [45], and others [47, 69, 70, 73, 39, 71]. DLPacker, the first deep learning-based model, employs a 3D convolution network to learn the density map of side-chain atoms, but it lacks the ability to capture rotation equivariance due to its 3D-CNN structure. In contrast, AttnPacker, the current state-of-the-art model, directly predicts side-chain atom coordinates using Tensor Field Network and SE(3)-Transformer, ensuring rotation equivariance in side chain packing. However, it does not consider the restrictions imposed by covalent bond lengths and angles, leading to inefficient training and unnatural bond lengths during generation. Furthermore, previous methods that treat side-chain packing as a regression problem assume a single ground-truth side-chain structure and overlook the fact that proteins can fold into diverse structures under different environmental factors, resulting in a distribution of side-chain conformations.

In this study, we depart from the standard practice of focusing on atom-level coordinates in Cartesian space as in prior research [45, 46]. Instead, we introduce **DiffPack**, a torsional diffusion model that models the exact degree of freedom in side-chain packing, the joint distribution of four torsional angles. By perturbing and denoising in the torsional space, we use an SE(3)-invariant network to learn the gradient field for the joint distribution of torsional angles. This result in a much smaller conformation space of side-chain, thereby capturing the intricate energy landscape of protein side chains. Despite its effectiveness, a direct joint diffusion process on the four torsion angles could result in steric clashes and accumulative coordinate displacement, which complicates the denoising process. To address this, we propose an autoregressive diffusion process and train separate diffusion models to generate the four torsion angles from $\chi_1$ to $\chi_4$ in an autoregressive manner. During training, each diffusion model only requires perturbation on its corresponding torsional angle using a teacher-forcing strategy, preserving the protein structure and avoiding the aforementioned issues. To improve the capacity of our model, we further introduce three schemes in sampling for consistently improving the inference results: multi-round sampling, annealed temperature sampling, and confidence models.

We evaluate our method on several benchmarks for protein side-chain packing and compare it with existing state-of-the-art methods. Our results demonstrate that DiffPack outperforms existing state-of-the-art approaches, achieving remarkable improvements of $11.9\%$ and $13.5\%$ in angle accuracy on CASP13 and CASP14, respectively. Remarkably, these performance gains are achieved with a significantly smaller model size, approximately 60 times fewer parameters, highlighting the potential of autoregressive diffusion models in protein structure prediction. Furthermore, we showcase the effectiveness of our method in enhancing the accuracy of side-chain predictions in the AlphaFold2 model, indicating its complementary nature to existing approaches.

## 2 Background

**Protein.** Proteins are composed of a sequence of residues (amino acids), each containing an alpha carbon ($C_\alpha$) atom bounded to an amino group (-NH$_2$), a carboxyl group (-COOH) and a side-chain (-R) that identifies the residue type. Peptide bonds link consecutive residues through a dehydration synthesis process. The backbone of a protein consists of the $C\alpha$ atom and the connected nitrogen, carbon, and oxygen atoms. We use $\mathcal{S} = [s_1, s_2, ..., s_n]$ to denote the sequence of a protein with $n$ residues, where $s_i \in \{0, ..., 19\}$ denotes the type of the $i$-th residue.

**Protein Conformation.** Physical interactions between residues make a protein fold into its native 3D structure, *a.k.a.*, conformation, which determines its biologically functional activity. We use $\mathcal{X} = [x_1, x_2, ..., x_n]$ to denote the structure of a protein, where $x_i$ denotes the set of atom coordinates belonging to the $i$-th residue. The backbone structure $x_i^{(\text{bb})}$ of the $i$-th residue is a subset consisting of backbone atoms, *i.e.*, $C_\alpha$, N, C, O, while the side-chain consists of the remaining atoms $x_i^{(\text{sc})} = x_i \setminus x_i^{(\text{bb})}$. The backbone conformation $\mathcal{X}^{(\text{bb})}$ and side-chain conformation $\mathcal{X}^{(\text{sc})}$ of the protein are defined as the set of backbones and side-chains of all residues, respectively.

**Protein Side-Chain Packing.** Protein side-chain packing (PSCP) problem aims to predict the 3D coordinates $\mathcal{X}^{(\text{sc})}$ of side-chain atoms given backbone conformations $\mathcal{X}^{(\text{bb})}$ and protein sequence $\mathcal{S}$. That is, we aim to model the conditional distribution $p(\mathcal{X}^{(\text{sc})}|\mathcal{X}^{(\text{bb})}, \mathcal{S})$.

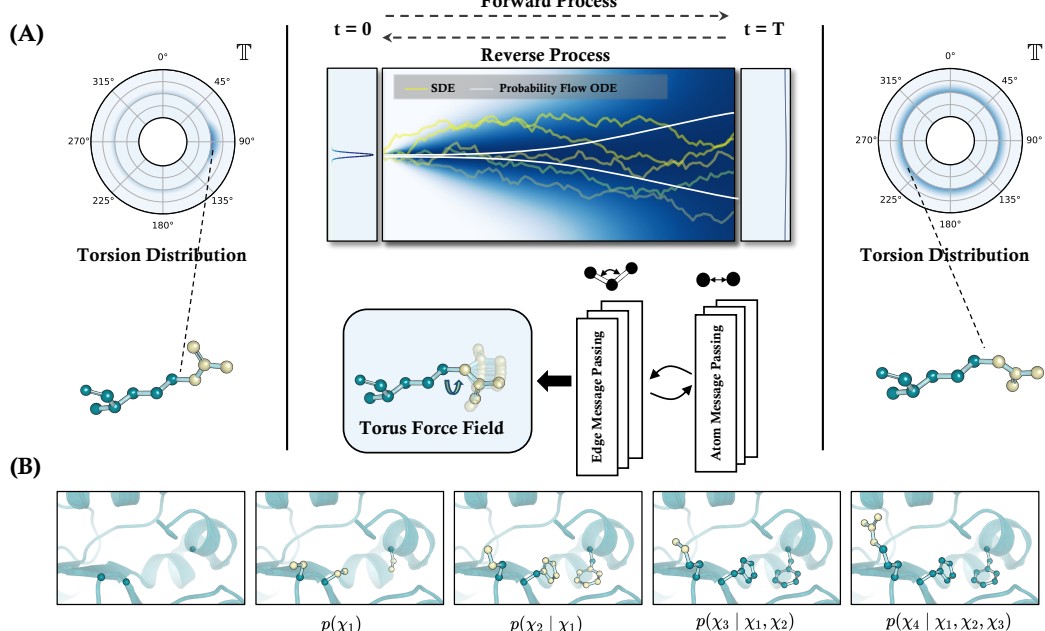

Figure 1: Overview of DiffPack. Given a protein sequence and backbone structure, we aim to model the conditional distribution of side-chain conformation. **(A)** Distribution of side-chain conformation is modeled through diffusion process in torsion space $\mathbb{T}^m$. An SE(3)-invariant network is used to learn the torus force field (torsion score). **(B)** Four torsion angles are generated autoregressively across all residues.

## 3 Methods

In this paper, we introduce an autoregressive diffusion model DiffPack to predict the side-chain conformation distribution in torsion space. We address the issue of overparameterization by introducing a torsion-based formulation of side-chain conformation in Section 3.1. Then we give a formulation of the torsional diffusion model in Section 3.2. However, directly applying the diffusion model encounters challenges in learning the joint distribution, which we address by introducing an autoregressive-style model in Section 3.3. We then provide details of the model architecture in Section 3.4, followed by an explanation of the inference procedure in Section 3.5.

### 3.1 Modeling Side-Chain Conformations with Torsional Angles

Previous methods [45, 46] model the side chain conformation as a series of three-dimensional coordinates in the Cartesian space. However, this approach does not take into account the restrictions imposed by constant covalent bond lengths and angles on the side chain's degree of freedom, resulting in inefficient training and unnatural bond lengths during generation.

To overcome this overparameterization issue, we propose modeling side chain conformation in torsion space. As illustrated in Figure 2, torsional angles directly dictate protein side-chain conformation by determining the twist between two neighboring planes. Table 6 lists the atom groups to define corresponding the neighboring plane for each residue type. The number of torsional angles varies across different residues, with a maximum of four ($\chi_1, \chi_2, \chi_3, \chi_4$). Modeling side chains in torsion space reduces the number of variables by approximately one third and restricts degrees of freedom [6]. Formally, we transform the problem of side-chain packing to modeling the distribution of torsional angles:

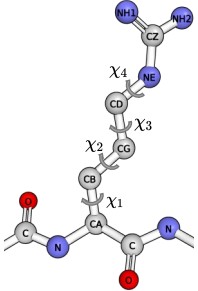

Figure 2: Illustration of four torsional angles.

$$p(\mathcal{X}^{(\text{sc})}|\mathcal{X}^{(\text{bb})}, \mathcal{S}) \Leftrightarrow p\left(\chi_1, \chi_2, \chi_3, \chi_4 | \mathcal{X}^{(\text{bb})}, \mathcal{S}\right), \tag{1}$$

where $\boldsymbol{\chi}_i \in [0, 2\pi)^{m_i}$ is a vector of the $i$-th torsional angles of all residues in the protein and we use $m_i$ to denote the number of residues with $\chi_i$. The space of possible side chain conformation is, therefore, reduced to an $m$-dimension sub-manifold $\mathcal{M} \subset \mathbb{R}^{3n}$ with $m = m_1 + m_2 + m_3 + m_4$.

## 3.2 Diffusion Models on Torsional Space

Denoising diffusion probabilistic models are generative models that learn the data distribution via a *forward diffusion process* and a *reverse generation process* [23]. We follow [30] to define diffusion models on the torsional space with the continuous score-based framework in [57]. For simplicity, we omit the condition and aggregate torsional angles on all residues as a torsional vector $\boldsymbol{\chi} \in [0, 2\pi)^m$. Starting with the data distribution as $p_0(\boldsymbol{\chi})$, the *forward diffusion process* is modeled by a stochastic differential equation (SDE):

$$d\boldsymbol{\chi} = \boldsymbol{f}(\boldsymbol{\chi}, t)\, dt + g(t)\, d\boldsymbol{w}, \tag{2}$$

where $\boldsymbol{w}$ is the Wiener process on the torsion space and $f(\boldsymbol{\chi}, t), g(t)$ are the drift coefficient and diffusion coefficient, respectively. Here we adopt Variance-Exploding SDE where $f(\boldsymbol{\chi}, t) = 0$ and $g(t)$ is exponentially decayed with $t$. With sufficiently large $T$, the distribution $p_T(\boldsymbol{\chi})$ approaches a uniform distribution over the torsion space. The *reverse generation process* samples from the prior and generates samples from the data distribution $p_0(\boldsymbol{\chi})$ via approximately solving the reverse SDE:

$$d\boldsymbol{\chi} = \left[\boldsymbol{f}(\boldsymbol{\chi}_t, t) - g^2(t)\nabla_{\boldsymbol{\chi}} \log p_t(\boldsymbol{\chi}_t)\right] dt + g(t)\, d\boldsymbol{w}, \tag{3}$$

where a neural network is learned to fit the score $\nabla_{\boldsymbol{\chi}} \log p_t(\boldsymbol{\chi}_t)$ of the diffused data [23, 57]. Inspired by [30], we convert the torsional angles into 3D atom coordinates and define the score network on Euclidean space, enabling it to explicitly learn the interatomic interactions.

The training process involves sampling from the perturbation kernel of the forward diffusion and computing its score to train the score network. Given the equivalence $(\chi_1, ..., \chi_m) \sim (\chi_1 + 2\pi, ..., \chi_m)... \sim (\chi_1, ..., \chi_m + 2\pi)$ in torsion space, the perturbation kernel is a wrapped Gaussian distribution on $\mathbb{R}^m$. This means that any $\boldsymbol{\chi}, \boldsymbol{\chi}' \in [0, 2\pi)^m$, the perturbation kernel is proportional to the sum of exponential terms, which depends on the distance between $\boldsymbol{\chi}$ and $\boldsymbol{\chi}'$:

$$p_{t|0}(\boldsymbol{\chi}'|\boldsymbol{\chi}) \propto \sum_{\boldsymbol{d} \in \mathbb{Z}^m} \exp\left(-\frac{\|\boldsymbol{\chi} - \boldsymbol{\chi}' + 2\pi\boldsymbol{d}\|^2}{2\sigma^2(t)}\right), \tag{4}$$

In order to sample from the perturbation kernel, we sample from the corresponding unwrapped isotropic normal and take element-wise $\mod 2\pi$. The kernel's scores are pre-computed using a numerical approximation. During training, we uniformly sampled a time step $t$ and the denoising score matching loss is minimized:

$$J_{\text{DSM}}(\theta) = \mathbb{E}_t\left[\lambda(t)\mathbb{E}_{\boldsymbol{\chi}_0 \sim p_0, \boldsymbol{\chi}_t \sim p_{t|0}(\cdot|\boldsymbol{\chi}_0)}\left[\|\boldsymbol{s}(\boldsymbol{\chi}_t, t) - \nabla_t \log p_{t|0}(\boldsymbol{\chi}_t|\boldsymbol{\chi}_0)\|^2\right]\right], \tag{5}$$

where the weight factors $\lambda(t) = 1/\mathbb{E}_{\boldsymbol{\chi} \sim p_{t|0}(\cdot|0)}\left[\|\nabla_{\boldsymbol{\chi}} \log p_{t|0}(\boldsymbol{\chi}|\boldsymbol{0})\|^2\right]$ are also pre-computed.

## 3.3 Autoregressive Diffusion Models

The direct approach to torsional diffusion models introduces noise on all torsional angles simultaneously and poses two significant challenges:

**1. Cumulative coordinate displacement**: Adding noises to torsional angles is equivalent with rotating side-chain atoms in and beyond the corresponding atom groups. For instance, if $\chi_1$ is rotated, it affects the coordinates of atoms in the $\chi_2$, $\chi_3$, and $\chi_4$ groups. This cumulative effect complicates the denoising process of the latter three angles. Similar issues arise after rotating $\chi_2$ and $\chi_3$. The effect of rotating $\chi_1$ is illustrated in Figure 3.

**2. Excessive steric clash**: Adding noises to all torsional angles may damage protein structures and complicate the denoising process in our score network. In Appendix C, Figure 8 compares the number of atom clashes observed when using noise schemes in vanilla diffusion models and autoregressive-style ones to show the straightforward application of diffusion models results in significantly more steric clashes.

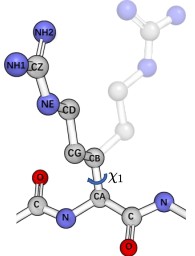

Figure 3: Effects of rotating $\chi_1$.

To address the aforementioned issues, we propose an autoregressive diffusion model over the torsion space. We factorize the joint distribution of the four torsional angles into separate conditional

distributions. Specifically, we have

$$p(\boldsymbol{\chi}_1, \boldsymbol{\chi}_2, \boldsymbol{\chi}_3, \boldsymbol{\chi}_4) = p(\boldsymbol{\chi}_1) \cdot p(\boldsymbol{\chi}_2 | \boldsymbol{\chi}_1) \cdot p(\boldsymbol{\chi}_3 | \boldsymbol{\chi}_{1,2}) \cdot p(\boldsymbol{\chi}_4 | \boldsymbol{\chi}_{1,2,3}). \qquad (6)$$

This allows us to model the side-chain packing problem as a step-by-step generation of torsional angles: first, we predict the first torsional angles $\boldsymbol{\chi}_1$ for all residues given the protein backbone structure; next, based on the backbone and the generated $\boldsymbol{\chi}_1$, we predict the second torsional angles $\boldsymbol{\chi}_2$; and so on for $\boldsymbol{\chi}_3$ and $\boldsymbol{\chi}_4$.

We train a separate score network for each distribution on the right-hand side using the torsional diffusion model from Section 3.2. To train the autoregressive model, we use a teacher-forcing strategy. Specifically, when modeling $p(\boldsymbol{\chi}_i | \boldsymbol{\chi}_{1,\dots,i-1})$, we assume that $\boldsymbol{\chi}_{1,\dots,i-1}$ are ground truth and keep these angles fixed. We then rotate $\boldsymbol{\chi}_i$ by sampling noises from the perturbation kernel in (4) and discard all atoms belonging to the $\boldsymbol{\chi}_{i+1,\dots,4}$ groups to remove the dependency on following torsional angles. This approach eliminates the cumulative effects of diffusion process on $\boldsymbol{\chi}_i$ and preserves the overall structure of the molecule, thus overcoming the aforementioned challenges. The generation process is illustrated in Figure 1 and the training procedure is described in Algorithm 1.

## 3.4 Model Architecture

To model $p(\boldsymbol{\chi}_i | \boldsymbol{\chi}_{1,\dots,i-1})$, we utilize a score network constructed using the 3D coordinates $\boldsymbol{s}(\mathcal{X}_t, t)$ of backbone atoms and atoms in the $\boldsymbol{\chi}_{1,\dots,i-1}$ groups. The score network's output represents the noise on torsional angles, which should be SE(3)-invariant *w.r.t.* the conformation $\mathcal{X}_t$. Unlike previous methods [45] operating in Cartesian coordinates that require an SE(3)-equivariant score network, our method only requires SE(3)-invariance, providing greater flexibility in designing the model architecture. To ensure this invariance, we employ GearNet-Edge [75], a state-of-the-art protein structure encoder. This involves constructing a multi-relational graph with atoms as nodes, where node features consist of one-hot encoding for atom types, corresponding residue types, and time step embeddings. Edges are added based on chemical bond and 3D spatial information, determining their type. To learn representations for each node, we perform relational message passing between them [50]. We denote the edge between nodes $i$ and $j$ with type $r$ as $(i, j, r)$ and set of relations as $\mathcal{R}$. We use $\boldsymbol{h}_i^{(l)}$ to denote the hidden representation of node $i$ at layer $l$. Then, message passing can be written as:

$$\boldsymbol{h}_i^{(l)} = \boldsymbol{h}_i^{(l-1)} + \sigma \left( \text{BatchNorm} \left( \sum_{r \in \mathcal{R}} \boldsymbol{W}_r \sum_{j \in \mathcal{N}_r(i)} \left( \boldsymbol{h}_j^{(l-1)} + \text{Linear} \left( \boldsymbol{m}_{(i,j,r)}^{(l)} \right) \right) \right) \right), \quad (7)$$

where $\boldsymbol{W}_r$ is the learnable weight matrix for relation type $r$, $\mathcal{N}_r(j)$ is the neighbor set of $j$ with relation type $r$, and $\sigma(\cdot)$ is the activation function. Edge representations $\boldsymbol{m}^{(l)}(i, j, r)$ are obtained through edge message passing. We use $e$ as the abbreviation of the edge $(i, j, r)$. Two edges $e_1$ and $e_2$ are connected if they share a common end node, with the connection type determined by the discretized angle between them. The edge message passing layer can be written as:

$$\boldsymbol{m}_{e_1}^{(l)} = \sigma \left( \text{BatchNorm} \left( \sum_{r \in \mathcal{R}'} \boldsymbol{W}_r' \sum_{e_2 \in \mathcal{N}_r'(e_1)} \boldsymbol{m}_{e_2}^{(l-1)} \right) \right), \qquad (8)$$

where $\mathcal{R}'$ is the set of relation types between edges and $\mathcal{N}_r'(e_1)$ is the neighbor set of $e_1$ with relation type $r$. After obtaining the hidden representations of all atoms at layer $L$, we compute the residue representations by taking the mean of the representations of its constituent atoms. The residue representations are then fed into an MLP for score prediction. The details of architectural components are summarized in Appendix D.

## 3.5 Inference

After completing the training, we adopt the common practice of autoregressive and diffusion models for inference, as described in Algorithm 2. We generate four torsional angles step by step as described in Section 3.3. When sampling $\boldsymbol{\chi}_i$ based on the predicted torsional angles $\boldsymbol{\chi}_{1,\dots,i-1}$, we begin by sampling a random angle from the uniform prior and then discretize and solve the reverse diffusion. At each time step, we generate atoms in the $\boldsymbol{\chi}_{1,\dots,i}$ group and use our learned score network for denoising. We also discover several simple techniques that significantly improve our performance, including multi-round sampling, annealed temperature sampling and confidence models.

**Predictor-corrector sampling.** After discretizing the reverse diffusion SDE, we perform multiple Langevin steps subsequent to each denoising step. This hybrid method, which mixes the denoising

process with Langevin dynamics, is called predictor-corrector sampling, as suggested by [57]. This approach can be seen as introducing an equilibration process to stabilize $p_t$, and it is demonstrated to be effective in diffusion process.

**Annealed temperature sampling.** When designing a generative model, two critical aspects to consider are quality and diversity. The diffusion model often suffers from overdispersion, which prioritizes diversity over sampling quality. However, in the context of side chain packing, quality is more important than diversity. Directly using the standard reverse sampling process may lead to undesirable structures. Following [28], we utilize an annealed temperature sampling scheme to mitigate this issue. Specifically, We modify the reverse SDE by adding an annealed weight $\lambda_t$ to the score function (details in Appendix E):

$$d\boldsymbol{\chi} = -\lambda_t \frac{d\sigma^2(t)}{dt} \nabla_{\boldsymbol{\chi}} \log p_t(\boldsymbol{\chi}_t)\, dt + \sqrt{\frac{d\sigma^2(t)}{dt}}\, d\boldsymbol{w}, \quad \text{where } \lambda_t = \frac{\sigma_{\max}^2}{T\sigma_{\max}^2 - (T-1)\sigma^2(t)}. \tag{9}$$

The above modification results in a reverse process approximating the low temperature sampling process, where ideally decreasing tempeture $T$ lead to a sampling process biased towards quality.

**Confidence model.** As per common practice in protein structure prediction [32], we train a confidence model to select the best prediction among multiple conformations sampled from our model. The model architecture we use is the same as that used in diffusion models, which takes the entire protein conformation as input and outputs representations for each residue. We train the model on the same dataset, using the residue representations to predict the negative residue-level RMSD of our sampled conformations. When testing, we rank all conformations based on our confidence model.

### 3.6 Handling Symmetric Issues

Torsional angles generally exhibit a periodicity of $2\pi$. However, certain rigid side-chain structures possess a $\pi$-rotation-symmetry, meaning that rotating the torsional angle by $\pi$ does not yield distinct physical structures, as demonstrated in Figure 4. For instance, in the case of a tyrosine residue, the phenolic portion of its side chain ensures that a $\pi$ rotation of its $\boldsymbol{\chi}_2$ torsion angles only alters the internal atom name without affecting the actual structure.

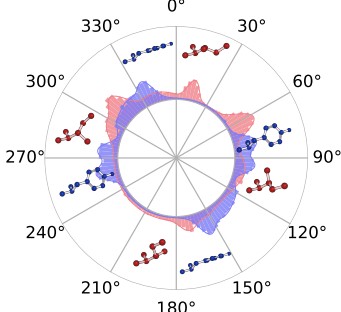

Figure 4: Distribution of $\pi$-rotation-symmetry torsional angles (Blue) and $2\pi$-rotation-symmetry (Red).

Previous research by Jumper et al. [32] addressed this concern by offering an alternative angle prediction, $\chi + \pi$, and minimizing the minimum distance between the ground-truth and both predictions. In our diffusion-based framework, we employ a distinct method. Specifically, the $\pi$-rotation-symmetry results in the equivalence $(\chi_i) \sim (\chi_i + k\pi)$ in torsion space, differing from the normal equivalence relationship in torsion space by a factor of $2k$. Consequently, we can still define the forwarding diffusion process in torsion space, albeit with a modification to Equation 4:

$$p_{t|0}(\boldsymbol{\chi}'|\boldsymbol{\chi}) \propto \sum_{\boldsymbol{d}\in\mathbb{Z}^m} \exp\left(-\frac{\|\boldsymbol{\chi}-\boldsymbol{\chi}'+\boldsymbol{c}\pi\boldsymbol{d}\|^2}{2\sigma^2(t)}\right), \quad \boldsymbol{c}\in\{1,2\}^m \tag{10}$$

where $\boldsymbol{c}_i = 1$ for $\pi$-rotation-symmetric rigid groups, and $\boldsymbol{c}_i = 2$ otherwise.

## 4 Related Work

**Protein side-chain packing.** Conventional approaches for protein side-chain packing (PSCP) involve minimizing the energy function over a pre-defined rotamer library [27, 73, 37, 3, 1, 35, 71, 7]. The choice of rotamer library, energy function, and energy minimization procedure varies among these methods. These methods rely on search heuristics and discrete sampling, limiting their accuracy. Currently, efficient methods like OSCAR-star [37], FASPR [27], SCWRL4 [35] do not incorporate deep learning and depend on rotamer libraries.

Several ML methods exist for side-chain prediction [47, 46, 69, 70, 73, 39, 71], including SIDE-Pro [47], which trains 156 feedforward networks to learn an additive energy function over pairwise atomic distances; DLPacker [46], which uses a deep U-net-style neural network to predict atom positions and selects the closest matching rotamer; OPUS-Rota4 [70], which employs multiple deep networks and utilizes MSA as input to predict side-chain coordinates and obtain a final structure;

and AttnPacker [45], which builds transformer layers and triangle updates based on components in Tensor Field Network [59] and SE(3)-Transformer [17] and achieves the state-of-the-art performance. In contrast, our method focuses solely on torsion space degrees of freedom and leverages an autoregressive diffusion model to accurately and efficiently model rotamer energy.

**Diffusion models on molecules and proteins.** The Diffusion Probabilistic Model (DPM), which was introduced in [55], has recently gained attention for its exceptional performance in generating images and waveforms [23, 10]. DPMs have been employed in a variety of problems in chemistry and biology, including molecule generation [72, 26, 68, 30], molecular representation learning [40], protein structure prediction [67], protein-ligand binding [12], protein design [2, 44, 28, 66, 38, 74], motif-scaffolding [60], and protein representation learning [76]. In this work, we investigate diffusion models in a new setting, protein side-chain packing, and propose a novel autoregressive diffusion model. Note that our definition of autoregressive diffusion model differs from existing works [25].

# 5  Experiments

## 5.1  Experimental Setup

**Dataset.** We use BC40 for training and validation, which BC40 is a subset of PDB database selected by $40\%$ sequence identity [63]. Following the dataset split in [45], there are 37266 protein chains for training and 1500 protein chains for validation. We evaluate our models on CASP13 and CASP14. Training set is curated so that no structure share sequence similarity with test set by $\geq 40\%$.

**Baselines.** We compare DiffPack with deep learning methods, like AttnPacker [45], DLPacker [46] and traditional methods including SCWRL4 [35], FASPR [27] and RosettaPacker [8]. Details can be found in Appendix H.1

**Metrics.** We evaluate the quality of generated side-chain conformations using three metrics: (1) **Angle MAE** measures the mean absolute error of predicted torsional angles. (2) **Angle Accuracy** measures the proportion of correct predictions, considering a torsional angle correct if the deviation is within $20°$. (3) **Atom RMSD** measures the average RMSD of side-chain atoms for each residue.

Since predicting surface side-chain conformations is considered more challenging, some results are divided into "**Core**" and "**Surface**" categories. Core residues are defined as those with at least 20 $C_\beta$ atoms within a 10Å radius, while surface residues have at most 15 $C_\beta$ atoms in the same region.

## 5.2  Side-Chain Packing

Table 1 summarizes the experimental result in CASP13, our model outperforms all other methods in predicting torsional angles, achieving the lowest mean absolute errors across all four Angle MAE categories ($\chi_1$, $\chi_2$, $\chi_3$, and $\chi_4$). Additionally, DiffPack shows the highest Angle Accuracy for all residues ($69.5\%$), core residues ($82.7\%$), and surface residues ($57.3\%$), where for surface residue the accuracy is increased by $20.4\%$ compared with previous state-of-the-art model AttnPacker. These results demonstrate that our model is better at capturing the distribution of the side chain torsion angles in both protein surfaces and cores. As for the atom-level side chain conformation prediction, DiffPack clearly outperforms other models in Atom RMSD. Moreover, the intrinsic design of DiffPack ensures that the generated structures have legal bond lengths and bond angles, while previous models in atom-coordinate space (e.g. AttnPacker) can easily generate side chains with illegal bond constraints without post-processing (as illustrated in Figure 6A).

Similarly, DiffPack outperforms other methods in all Angle MAE categories on the CASP14 dataset (Table 2). It achieves the highest Angle Accuracy for all residues ($57.5\%$), core residues ($77.8\%$), and surface residues ($43.5\%$). Furthermore, DiffPack reports the best Atom RMSD for all residues ($0.793$ Å), core residues ($0.356$ Å), and surface residues ($0.956$ Å).

Despite its superior performance on both test sets, our model, DiffPack, has a significantly smaller number of total parameters (3,043,363) compared to the previous state-of-the-art model, AttnPacker (208,098,163), which relies on multiple layers of complex triangle attention. This substantial reduction ($68.4\times$) in model size highlights the efficiency of diffusion-based approaches like DiffPack, making them more computationally feasible and scalable solutions for predicting side-chain conformations.

| Method | ANGLE MAE ° ↓ | | | | ANGLE ACCURACY % ↑ | | | ATOM RMSD Å ↓ | | |
|---|---|---|---|---|---|---|---|---|---|---|
| | $\chi_1$ | $\chi_2$ | $\chi_3$ | $\chi_4$ | All | Core | Surface | All | Core | Surface |
| SCWRL | 27.64 | 28.97 | 49.75 | 61.54 | 56.2% | 71.3% | 43.4% | 0.934 | 0.495 | 1.027 |
| FASPR | 27.04 | 28.41 | 50.30 | 60.89 | 56.4% | 70.3% | 43.6% | 0.910 | 0.502 | 1.002 |
| RosettaPacker | 25.88 | 28.25 | 48.13 | 59.82 | 58.6% | 75.3% | 35.7% | 0.872 | 0.422 | 1.001 |
| DLPacker | 22.18 | 27.00 | 51.22 | 70.04 | 58.8% | 73.9% | 45.4% | 0.772 | 0.402 | 0.876 |
| AttnPacker | 18.92 | 23.17 | 44.89 | 58.98 | 62.1% | 73.7% | 47.6% | 0.669 | 0.366 | 0.775 |
| DiffPack | **15.35** | **19.19** | **37.30** | **50.19** | **69.5%** | **82.7%** | **57.3%** | **0.579** | **0.298** | **0.696** |

Table 1: Comparative evaluation of DiffPack and prior methods on CASP13.

| Method | ANGLE MAE ° ↓ | | | | ANGLE ACCURACY % ↑ | | | ATOM RMSD Å ↓ | | |
|---|---|---|---|---|---|---|---|---|---|---|
| | $\chi_1$ | $\chi_2$ | $\chi_3$ | $\chi_4$ | All | Core | Surface | All | Core | Surface |
| SCWRL | 33.50 | 33.05 | 51.61 | 55.28 | 45.4% | 62.5% | 33.2% | 1.062 | 0.567 | 1.216 |
| FASPR | 33.04 | 32.49 | 50.15 | 54.82 | 46.3% | 62.4% | 34.0% | 1.048 | 0.594 | 1.205 |
| RosettaPacker | 31.79 | 28.25 | 50.54 | 56.16 | 47.5% | 67.2% | 33.5% | 1.006 | 0.501 | 1.183 |
| DLPacker | 29.01 | 33.00 | 53.98 | 72.88 | 48.0% | 66.9% | 33.9% | 0.929 | 0.476 | 1.107 |
| AttnPacker | 25.34 | 28.19 | 48.77 | **51.92** | 50.9% | 66.2% | 36.3% | 0.823 | 0.438 | 1.001 |
| DiffPack | **21.91** | **25.54** | **44.27** | 55.03 | **57.5%** | **77.8%** | **43.5%** | **0.770** | **0.356** | **0.956** |

Table 2: Comparative evaluation of DiffPack and prior methods on CASP14.

## 5.3 Side-Chain Packing on Non-Native Backbone

In addition to native backbones, another interesting and challenging problem is whether the side chain packing algorithm can be applied to non-native backbone (*e.g.*, backbones generated by protein folding algorithms). In this regard, we extend DiffPack to accommodate non-native backbones generated from AlphaFold2 [32]. Table 3 gives the quantitative result of different algorithms including AlphaFold2's side-chain prediction on the CASP13-FM test set. All metrics are calculated after aligning the non-native backbone of each residue to the native backbone. As observed, DiffPack achieves state-of-the-art on most metrics. Notably, DiffPack is the only model that consistently outperforms AlphaFold2 in all metrics, showcasing its potential to refine AlphaFold2's predictions.

## 5.4 Ablation Study

To analyze the contribution of different parts in our proposed method, we perform ablation studies on CASP13 and CASP14 benchmarks. The results are shown in Table 4.

**Autoregressive diffusion modeling.** We evaluate our autoregressive diffusion modeling approach against two baselines: joint diffusion ($\chi_{1,2,3,4}$) and random diffusion ($\chi_i$). Joint diffusion models the joint distribution of the four torsional angles of each residue and performed diffusion and denoising on all angles simultaneously, while random diffusion perturbs one random torsional angle per residue and generates all angles simultaneously during inference. Our approach outperforms both baselines (Table 4). Training loss curves (Figure 5) show that random diffusion has an easier time optimizing its loss than joint diffusion, but struggles with denoising all angles at once due to a mismatch between training and inference objectives. Our autoregressive scheme strikes a balance between ease of training and quality of generation, achieving good performance.

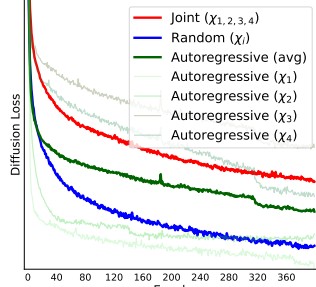

Figure 5: Training loss curves for different diffusion models.

We plot the training curves of the conditional distributions for the four torsional angles in DiffPack, finding that $\chi_1$ and $\chi_2$ are easier to optimize and perform better than random diffusion, likely due to discarding atoms from subsequent angles to overcome cumulative perturbation effects. However, training on the smaller set of valid residues with $\chi_3$ and $\chi_4$ is inefficient. Future work should address the challenge of training these angles more efficiently and mitigating cumulative errors in autoregressive models.

**Inference.** Three techniques are proposed in Section 3.5 to improve the inference of DiffPack. To evaluate the effectiveness of these techniques, we compare them with three baselines that do not

| | ANGLE MAE $^\circ$ $\downarrow$ | | | | ANGLE ACCURACY % $\uparrow$ | | | ATOM RMSD Å $\downarrow$ | | |
|---|---|---|---|---|---|---|---|---|---|---|
| **Method** | $\chi_1$ | $\chi_2$ | $\chi_3$ | $\chi_4$ | All | Core | Surface | All | Core | Surface |
| AlphaFold2* | 35.20 | 31.10 | 51.38 | 56.95 | 51.3% | 71.5% | 38.7% | 1.058 | 0.521 | 1.118 |
| SCWRL | 34.94 | 30.84 | 50.45 | 56.75 | 51.3% | 69.5% | 39.0% | 1.079 | 0.550 | 1.148 |
| FASPR | 34.83 | 30.85 | 50.60 | 56.74 | 50.8% | 67.9% | 39.8% | 1.073 | 0.573 | 1.114 |
| RosettaPacker | 35.43 | 31.63 | 51.33 | **56.18** | 50.9% | 70.6% | 38.5% | 1.070 | 0.526 | 1.139 |
| DLPacker | 34.38 | 31.57 | 55.84 | 67.02 | 49.9% | 69.1% | 37.0% | 1.032 | 0.543 | 1.090 |
| AttnPacker | 33.23 | 31.97 | 50.53 | 58.20 | 51.0% | 68.4% | 39.1% | 0.981 | 0.512 | **1.027** |
| DiffPack | **31.25** | **30.17** | **48.32** | 56.82 | **55.5%** | **74.3%** | **41.9%** | **0.978** | **0.490** | 1.056 |

Table 3: Comparative evaluation on CASP13-FM non-native backbones generated by AlphaFold2.

| | CASP13 ANGLE MAE $^\circ$ $\downarrow$ | | | | CASP14 ANGLE MAE $^\circ$ $\downarrow$ | | | |
|---|---|---|---|---|---|---|---|---|
| **Method** | $\chi_1$ | $\chi_2$ | $\chi_3$ | $\chi_4$ | $\chi_1$ | $\chi_2$ | $\chi_3$ | $\chi_4$ |
| DiffPack | **15.35** | **19.19** | 37.30 | 50.19 | **21.91** | **25.54** | **44.27** | 55.03 |
| - w/ joint diffusion ($\chi_{1,2,3,4}$) | 17.14 | 23.72 | **35.96** | **45.30** | 26.80 | 34.51 | 52.77 | 63.41 |
| - w/ random diffusion ($\chi_i$) | 17.11 | 27.27 | 44.75 | 56.42 | 23.26 | 32.69 | 49.21 | **51.92** |
| - w/o multi-round sampling | 16.26 | 23.13 | 40.60 | 52.67 | 23.86 | 30.71 | 47.54 | 55.80 |
| - w/o annealed temperature | 15.55 | 22.56 | 39.32 | 51.37 | 22.82 | 29.61 | 45.86 | 55.22 |
| - w/o confidence models | 16.12 | 22.92 | 39.92 | 50.63 | 22.86 | 29.30 | 45.80 | 54.08 |

Table 4: Ablation study on CASP13 and CASP14.

use these techniques. For the baselines without multi-round sampling and annealed temperature, we simply resume the sampling procedure in the vanilla diffusion models. For the baseline without confidence models, we only draw one sample from our model instead of using confidence models to ensemble multiple samples. As shown in Table 4, the mean absolute error of the four torsional angles increases for the three baselines, demonstrating the effectiveness of our proposed techniques.

## 5.5 Case Study

**DiffPack accurately predict the side-chain conformation with chemical validity.** As shown in Figure 6A and Figure 6B. DiffPack accurately predict the side-chain conformation with a substantially lower RMSD (0.196Å and 0.241Å) compared with other deep learning methods. Furthermore, DiffPack consistently ensures the validity of generated structures, while AttnPacker without post-processing sometimes violates the chemical validity due to its operation on atom coordinates.

**DiffPack correctly identifies the $\pi$ stacking interaction.** Accurate reconstruction of $\pi$ stacking interaction between side-chain has traditionally been challenging. Traditional method usually requires a specific energy term for modeling this interaction. Interestingly, DiffPack has shown the ability to implicitly model this interaction without the need of additional prior knowledge (Figure 6C).

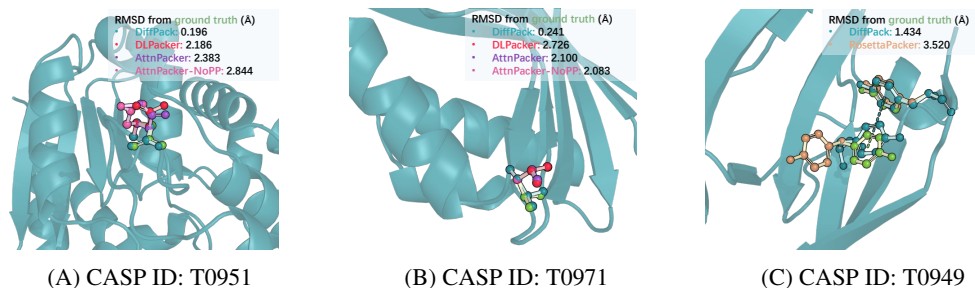

(A) CASP ID: T0951      (B) CASP ID: T0971      (C) CASP ID: T0949

Figure 6: Case studies on DiffPack. Predictions from different methods are distinguished by color. **(A)** DiffPack accurately predicts the side-chain conformation. AttnPacker-NoPP produces an invalid glutamic acid structure since $O^{\delta 1}$ is too close to $O^{\delta 2}$. **(B)** DiffPack accurately predicts the $\chi_1$ of leucine. **(C)** DiffPack correctly identifies $\pi$-$\pi$ stacking interactions, indicated by dashed lines.

## 6 Conclusions

In this paper, we present DiffPack, a novel approach that models protein side-chain packing using a diffusion process in torsion space. Unlike vanilla joint diffusion processes, DiffPack incorporates an autoregressive diffusion process, addressing certain limitations. Our empirical results demonstrate the superiority of our proposed method in predicting protein side-chain conformations compared to existing approaches. Future directions include exploring diffusion processes for sequence and side-chain conformation co-generation, optimizing computational efficiency in side-chain packing, and considering backbone flexibility in the diffusion process.

## Acknowledgement

We would like to thank Ramanarayan Mohanty from Intel for his support in optimizing the speed of DiffPack. We also extend our appreciation to Zhaocheng Zhu, Sophie Xhonneux and Chuanrui Wang for their constructive feedback on the manuscript. This project is supported by Twitter, Intel, the Natural Sciences and Engineering Research Council (NSERC) Discovery Grant, the Canada CIFAR AI Chair Program, Samsung Electronics Co., Ltd., Amazon Faculty Research Award, Tencent AI Lab Rhino-Bird Gift Fund, a NRC Collaborative R&D Project (AI4D-CORE-06) as well as the IVADO Fundamental Research Project grant PRF-2019-3583139727.

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

# A   More Related Work

**Geometric deep learning on biomolecules.**   Learning protein representations based on 3D geometric information is crucial for various protein tasks. Recent advancements have led researchers to develop architectures that preserve properties such as invariance and equivariance for essential transformations like rotation and translation. These approaches have utilized various techniques such as node/atom message passing[19, 51, 52, 49], edge/bond message passing [31, 9], and directional information [33, 41, 34] to encode molecular graphs in 2D or 3D. Notably, recent models have been generalized to protein 3D structures [20, 5, 29, 22, 21] and protein surfaces [18, 58, 13, 56], demonstrating remarkable performance on diverse tasks. In this work, we utilize a state-of-the-art protein structure encoder, GearNet-Edge [75] to obtain SE(3)-invariant representations for denoising in the torsion space.

## A.1   Broader Impact

The impact of our work extends well beyond the realm of protein structure prediction, with implications across various fields of biological science and medical research. The ability to accurately predict protein side-chain conformations is of vital importance in understanding the intricate functioning of proteins and their interactions with other molecules, including substrates, inhibitors, and drugs.

**Drug Design and Discovery:** Accurate prediction of side-chain conformations could considerably advance the field of drug discovery and design, where the interaction between proteins and small molecule drugs or inhibitors is often determined by the precise conformation of side chains. Improved accuracy in side-chain packing can facilitate more precise predictions of protein-ligand binding sites and affinities, thereby expediting the development of new therapeutics.

**Protein Engineering:** Effective application of our method could also revolutionize protein engineering, where side-chain packing plays a pivotal role in protein stability, function, and interaction. Enhancing our understanding of side-chain packing will help engineer proteins with desired properties, including altered substrate specificity, increased stability, or novel functionality.

**Disease Understanding:** Many diseases, such as Alzheimer's, Parkinson's, and various cancers, are tied to misfolded proteins or mutations that affect protein structure. Accurate side-chain prediction can thus provide key insights into the structural consequences of such mutations, supporting the development of targeted treatments.

**Bioinformatics and Computational Biology:** On a technical note, our work provides a novel perspective for side-chain packing problem by formulating it as a diffusion process in the torsional space, which could inspire further innovations in bioinformatics and computational biology. Moreover, given the efficiency of our approach, it is anticipated to facilitate the rapid and scalable analysis of large-scale protein datasets, aiding in the exploration of the proteome.

## A.2   Limitation

In this study, we focus exclusively on protein side-chain prediction under the assumption of a fixed and highly accurate backbone. Nevertheless, in real-world scenarios, protein backbones may be generated using existing structure prediction methods, which may not provide sufficient accuracy. It is essential to address how our methods can be adapted to accommodate such settings, and this aspect remains an important future direction for our research. Additionally, we intend to explore the application of our methods in various downstream tasks, such as protein-protein interactions and protein engineering tasks, as part of our future research directions.

# B   Additional Experiment Results

## B.1   Steric Clash

In order to quantify whether the generated structure has severe steric clash, we add an additional metric **Steric Clash Count** measures the mean number of clash pairs appearing in generated structures. Following previous work [45, 36], the distance threshold between different types of atoms is initially defined by van der Waals radii and further adjusted to account for factors such as H-bond and disulfide

bridges. A protein atom pair is identified to have a steric clash if the inter-atomic distance is within X% of the threshold. Here, X is chosen as $100\%$, $90\%$, and $80\%$, representing varying levels of stringency and providing a more comprehensive assessment of steric clashes.

| Method | CASP13 CLASH COUNT ↓ | | | CASP14 CLASH COUNT ↓ | | |
|---|---|---|---|---|---|---|
| | 100% | 90% | 80% | 100% | 90% | 80% |
| SCWRL | 115.3 | 20.6 | 4.6 | 124.0 | 24.6 | 6.5 |
| FASPR | 112.8 | 23.3 | 5.6 | 130.5 | 29.5 | 8.7 |
| RosettaPacker | 73.8 | 7.9 | 2.6 | 100.6 | 10.1 | 3.4 |
| DLPacker | 64.3 | 7.3 | 2.0 | 74.1 | 10.5 | 3.0 |
| AttnPacker-noPP | 40.1 | 5.7 | 1.5 | 49.5 | 10.2 | 3.5 |
| DiffPack | **37.5** | **4.6** | **0.9** | **46.5** | **6.0** | **1.1** |

Table 5: Mean Clash Pair Number of Generation Result

From Table 5, it is clear that the DiffPack method outperforms the other techniques across all levels of stringency for both CASP13 and CASP14 datasets. This superiority suggests that the DiffPack method produces protein structures with fewer steric clashes, indicating a more realistic and physically plausible model of protein side chain packing.

At the 100% distance threshold, DiffPack generates 67.5% fewer clashes than SCWRL, 66.7% fewer than FASPR, 49.1% fewer than RosettaPacker, and 41.7% fewer than DLPacker for the CASP13 dataset. Similarly, for the CASP14 dataset, it generates 62.5% fewer clashes than SCWRL, 64.3% fewer than FASPR, 53.8% fewer than RosettaPacker, and 37.2% fewer than DLPacker.

At the 90% and 80% distance thresholds, the comparative reduction in clashes by DiffPack is also evident. This trend suggests that DiffPack consistently generates structures with fewer steric clashes, which is crucial for generating biologically feasible protein structures.

Interestingly, the AttnPacker-noPP method also shows a considerable reduction in clashes compared to other methods, especially SCWRL and FASPR, but it is still surpassed by the performance of DiffPack. This suggests that the autoregressive diffusion model used in DiffPack is more capable of managing steric clashes in protein structure generation, demonstrating the effectiveness of this approach.

## B.2 Visualization of Sampling Results

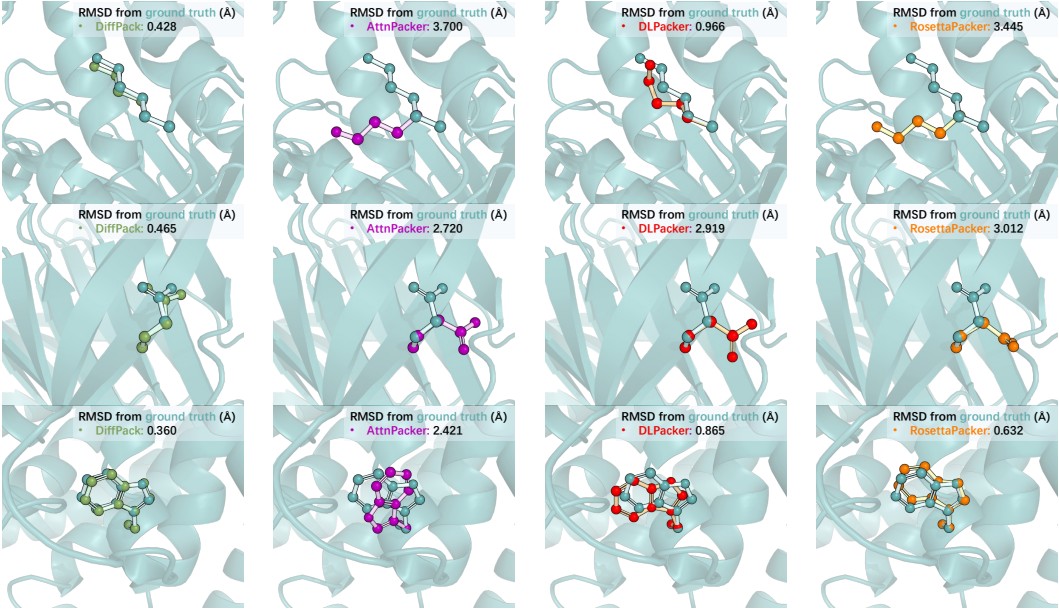

Figure 7: Visualization of sampling results. **DiffPack** (green), **AttnPacker** (purple), **DLPacker** (red), **RosettaPacker** (orange) are evaluated on T1000 (top), T0954 (middle), T1020 (bottom) cases.

# C Details of Autoregressive Diffusion Models

## C.1 Atom groups for each residue type

For completeness, we provide the definition of torsional angles and corresponding atom groups for each residue in Table 6. These definitions align with those presented in the AlphaFold2 paper [32].

| Residue Type | $\chi_1$ | $\chi_2$ | $\chi_3$ | $\chi_4$ |
|---|---|---|---|---|
| **ALA** | - | - | - | - |
| **ARG** | N, C$^\alpha$, C$^\beta$, C$^\gamma$ | C$^\alpha$, C$^\beta$, C$^\gamma$, C$^\delta$ | C$^\beta$, C$^\gamma$, C$^\delta$, N$^\epsilon$ | C$^\gamma$, C$^\delta$, N$^\epsilon$, C$^\zeta$ |
| **ASN** | N, C$^\alpha$, C$^\beta$, C$^\gamma$ | C$^\alpha$, C$^\beta$, C$^\gamma$, O$^{\delta 1}$ | - | - |
| **ASP** | N, C$^\alpha$, C$^\beta$, C$^\gamma$ | C$^\alpha$, C$^\beta$, C$^\gamma$, O$^{\delta 1}$ | - | - |
| **CYS** | N, C$^\alpha$, C$^\beta$, S$^\gamma$ | - | - | - |
| **GLN** | N, C$^\alpha$, C$^\beta$, C$^\gamma$ | C$^\alpha$, C$^\beta$, C$^\gamma$, C$^\delta$ | C$^\beta$, C$^\gamma$, C$^\delta$, O$^{\epsilon 1}$ | - |
| **GLU** | N, C$^\alpha$, C$^\beta$, C$^\gamma$ | C$^\alpha$, C$^\beta$, C$^\gamma$, C$^\delta$ | C$^\beta$, C$^\gamma$, C$^\delta$, O$^{\epsilon 1}$ | - |
| **GLY** | - | - | - | - |
| **HIS** | N, C$^\alpha$, C$^\beta$, C$^\gamma$ | C$^\alpha$, C$^\beta$, C$^\gamma$, N$^{\delta 1}$ | - | - |
| **ILE** | N, C$^\alpha$, C$^\beta$, C$^{\gamma 1}$ | C$^\alpha$, C$^\beta$, C$^{\gamma 1}$, C$^{\delta 1}$ | - | - |
| **LEU** | N, C$^\alpha$, C$^\beta$, C$^\gamma$ | C$^\alpha$, C$^\beta$, C$^\gamma$, C$^{\delta 1}$ | - | - |
| **LYS** | N, C$^\alpha$, C$^\beta$, C$^\gamma$ | C$^\alpha$, C$^\beta$, C$^\gamma$, C$^\delta$ | C$^\beta$, C$^\gamma$, C$^\delta$, C$^\epsilon$ | C$^\gamma$, C$^\delta$, C$^\epsilon$, N$^\zeta$ |
| **MET** | N, C$^\alpha$, C$^\beta$, C$^\gamma$ | C$^\alpha$, C$^\beta$, C$^\gamma$, S$^\delta$ | C$^\beta$, C$^\gamma$, S$^\delta$, C$^\epsilon$ | - |
| **PHE** | N, C$^\alpha$, C$^\beta$, C$^\gamma$ | C$^\alpha$, C$^\beta$, C$^\gamma$, C$^{\delta 1}$ | - | - |
| **PRO** | N, C$^\alpha$, C$^\beta$, C$^\gamma$ | C$^\alpha$, C$^\beta$, C$^\gamma$, C$^\delta$ | - | - |
| **SER** | N, C$^\alpha$, C$^\beta$, O$^\gamma$ | - | - | - |
| **THR** | N, C$^\alpha$, C$^\beta$, O$^{\gamma 1}$ | - | - | - |
| **TRP** | N, C$^\alpha$, C$^\beta$, C$^\gamma$ | C$^\alpha$, C$^\beta$, C$^\gamma$, C$^{\delta 1}$ | - | - |
| **TYR** | N, C$^\alpha$, C$^\beta$, C$^\gamma$ | C$^\alpha$, C$^\beta$, C$^\gamma$, C$^{\delta 1}$ | - | - |
| **VAL** | N, C$^\alpha$, C$^\beta$, C$^{\gamma 1}$ | - | - | - |

Table 6: Specification of atom groups defining the torsion angles ($\chi_1$, $\chi_2$, $\chi_3$, $\chi_4$) for each residue type. Torsion angle exhibiting $\pi$-rotation-symmetry is colored as purple. Other side-chain torsion angles exhibit $2\pi$-periodicity.

## C.2 Comparison between numbers of atom clashes of joint and autoregressive diffusion

In Figure 8, we provide a comparative illustration of atom clashes in protein structures using different noise schemes in autoregressive diffusion and joint diffusion process.

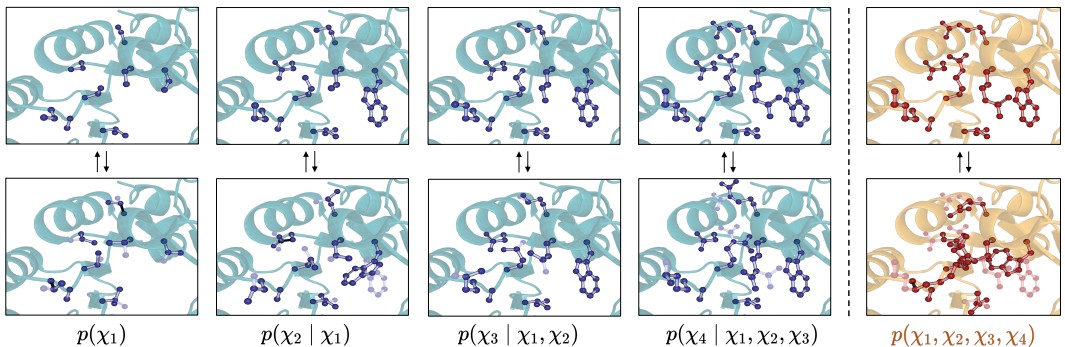

$p(\chi_1)$    $p(\chi_2 \mid \chi_1)$    $p(\chi_3 \mid \chi_1, \chi_2)$    $p(\chi_4 \mid \chi_1, \chi_2, \chi_3)$    $p(\chi_1, \chi_2, \chi_3, \chi_4)$

Figure 8: Comparative illustration of atom clashes in protein structures using noise schemes from autoregressive diffusion process (left) and vanilla joint diffusion process (right). Vanila joint diffusion process add noise to all torsion angles simultaneously, resulting in much more steric clash.

## C.3 Algorithm illustration of DiffPack

---

**Algorithm 1** Training Procedure

---

**Input:** training dataset $\mathcal{D}$ with protein sequence and structure pairs $(\mathcal{S}, \mathcal{X})$, learning rate $\alpha$

**Output:** trained score networks $s_{\theta_1}^{(1)}, s_{\theta_2}^{(2)}, s_{\theta_3}^{(3)}, s_{\theta_4}^{(4)}$ for torsional angles $\chi_1, \chi_2, \chi_3, \chi_4$

 1: **for** $epoch \leftarrow 1$ to $epoch_{\max}$ **do**
 2:      **for** $(\mathcal{S}, \mathcal{X})$ in $\mathcal{D}$ **do**
 3:          calculate torsional angles $\chi_1, \chi_2, \chi_3, \chi_4$ from structure $\mathcal{X}$;
 4:          **for** $i \leftarrow 1$ to $4$ **do**                     ▷ Train each model separately
 5:             sample $t \in [0, 1]$;
 6:             sample $\Delta\chi_i$ from wrapped normal $p_{t|0}(\cdot|0)$ with $\sigma = \sigma_{\min}^{1-t}\sigma_{\max}^t$;
 7:             set $\widetilde{\chi_i} = \chi_i + \Delta\chi_i$;
 8:             generate perturbed structure $\tilde{\mathcal{X}}$ with $\chi_{1..i-1}$ and $\widetilde{\chi_i}$ and discard atoms in $\chi_{i+1..4}$ groups;
 9:             predict $\delta\chi_i = s_{\theta_i}^{(i)}(\tilde{\mathcal{X}}, t)$;
10:             update $\theta_i \leftarrow \theta_i - \alpha\nabla_{\theta_i}\|\delta\chi_i - \nabla_{\Delta\chi_i}p_{t|0}(\Delta\chi_i|0)\|$;
11:          **end for**
12:      **end for**
13: **end for**

---

---

**Algorithm 2** Inference Procedure

---

**Input:** protein sequence $\mathcal{S}$ and backbone conformation $\mathcal{X}^{(\text{bb})}$, number steps $N$, number rounds $R$

**Output:** protein conformation $\mathcal{X}$

 1: **for** $i \leftarrow 1$ to $4$ **do**
 2:      sample $\chi_i \sim U[0, 2\pi]^{m_i}$;                     ▷ Initialize with uniform prior
 3:      generate structure $\tilde{\mathcal{X}}$ with $\chi_{1..i}$ and discard atoms in $\chi_{i+1..4}$ groups;
 4:      **for** $n \leftarrow N$ to $1$ **do**
 5:          let $t = n/N, g(t) = \sigma_{\min}^{1-t}\sigma_{\max}^t\sqrt{2\ln(\sigma_{\max}/\sigma_{\min})}$;
 6:          **for** $r \leftarrow 1$ to $R$ **do**                   ▷ Multi-round sampling
 7:             predict $\delta\chi_i = s_{\theta_i}^{(i)}(\tilde{\mathcal{X}}, t)$;
 8:             draw $z$ from wrapped normal with $\sigma^2 = 1/N$;
 9:             set $\Delta\chi_i = \lambda_t(g(t)^2/N)\delta\chi_i + g(t)z$;
10:             update $\chi_i \leftarrow \chi_i + \Delta\chi_i$;               ▷ Update torsional angles
11:             generate structure $\tilde{\mathcal{X}}$ with $\chi_{1...i}$ and discard atoms in $\chi_{i+1..4}$ groups;
12:          **end for**
13:      **end for**
14: **end for**
15: **return** structure $\mathcal{X}$ generated with $\chi_1, \chi_2, \chi_3, \chi_4$

---

# D   Score Network Architecture

**Atom graph construction.** We represent protein structure as an atom-level relational graph, where atoms serve as nodes and are connected by edges based on three conditions: (1) if they are linked by a chemical bond, (2) if one is among the 10-nearest neighbors of the other, and (3) if their Euclidean distance is within $4.5\text{Å}$ and their sequential distance is above $2$. Each type of edge is treated as a different relation, while the node feature is created by concatenating the one-hot features of atom and residue types. Following [75], edge features are created by concatenating one-hot features of residue types, relation types, and sequential and Euclidean distances. This graph construction considers both geometric and sequential properties of proteins, providing a comprehensive featurization of proteins.

**Embedding layer.** In the embedding layer, we fuse node features with embeddings for encoding time steps. We use sinusoidal embeddings [61] of time $t$ as input for a linear layer. Node features are passed through a Multi-Layer Perceptron (MLP) and added with time embeddings to obtain the new node features for the model.

**Message passing layer.** To learn representations for each node, we perform relational message passing between them [50]. We denote the edge between nodes $i$ and $j$ with type $r$ as $(i, j, r)$ and set

of relations as $\mathcal{R}$. We use $\boldsymbol{h}_i^{(l)}$ to denote the hidden representation of node $i$ at layer $l$. Then, message passing can be written as:

$$\boldsymbol{h}_i^{(l)} = \boldsymbol{h}_i^{(l-1)} + \sigma \left( \text{BN} \left( \sum_{r \in \mathcal{R}} \boldsymbol{W}_r \sum_{j \in \mathcal{N}_r(i)} \left( \boldsymbol{h}_j^{(l-1)} + \text{Linear} \left( \boldsymbol{m}_{(i,j,r)}^{(l)} \right) \right) \right) \right), \quad (11)$$

where $\boldsymbol{W}_r$ is the learnable weight matrix for relation type $r$, $\mathcal{N}_r(j)$ is the neighbor set of $j$ with relation type $r$, $\text{BN}(\cdot)$ denotes batch normalization, and $\sigma(\cdot)$ is the ReLU function. We use $\boldsymbol{m}^{(l)}(i, j, r)$ to denote the representation of edge $(i, j, r)$, computed through edge message passing. We use $e$ as the abbreviation of the edge $(i, j, r)$. Two edges $e_1$ and $e_2$ are connected if they share a common end node. The type of their connection is determined by the angle between them, which is discretized into 8 bins. The edge message passing layer can be written as:

$$\boldsymbol{m}_{e_1}^{(l)} = \sigma \left( \text{BN} \left( \sum_{r \in \mathcal{R}'} \boldsymbol{W}_r' \sum_{e_2 \in \mathcal{N}_r'(e_1)} \boldsymbol{m}_{e_2}^{(l-1)} \right) \right), \quad (12)$$

where $\mathcal{R}'$ is the set of relation types between edges and $\mathcal{N}_r'(e_1)$ is the neighbor set of $e_1$ with relation type $r$. After obtaining the hidden representations of all atoms at layer $L$, we compute the residue representations by taking the mean of the representations of its constituent atoms. Finally, we apply an MLP to the residue representations to predict the score.

## E  Annealed Sampling

As discussed in Section 3.5, vanilla diffusion sampling scheme suffers from over-dispersion problem, which have negative influence in side-chain packing. A promising way to tackle this issue is low-temperature sampling, which involves perturbing the initial distribution $p(x)$ with a temperature factor $T$. This results in a re-normalized distribution $p_T(\mathbf{x}) = \frac{1}{Z} p(\mathbf{x})^{\frac{1}{T}}$. Lowering the temperature value shifts the model's focus towards sampling quality, while increasing it emphasizes diversity. However, strict low temperature sampling is an expensive iterative process. Merely up-scaling the score function or down-scaling the noise term in the reverse SDE does not resolve this challenge.

Inspired from [28], we proposed a modified reverse SDE to approximate the low-temperature sampling. Specifically, the modified reverse SDE is first derived by considering a simplified Gaussian data distribution $\mathcal{N}\left(\mathbf{x}_0; \boldsymbol{\mu}_{\text{data}}, \sigma_{\text{data}}^2\right)$. In this simplified case, we show that the modified reverse SDE with the annealed weight $\lambda_t$ converges to a low-temperature sampling process that prioritizes quality while maintaining diversity.

Considering the simplified data distribution $\mathcal{N}\left(\mathbf{x}_0; \boldsymbol{\mu}_{\text{data}}, \sigma_{\text{data}}^2\right)$, the time-dependent marginal density of noise-perturbed distribution by applying VE-SDE forwarding process $d\mathbf{x} = \sqrt{\frac{d\sigma^2(t)}{dt}} d\boldsymbol{w}$ is

$$p_t(\mathbf{x}) = \mathcal{N}\left(\mathbf{x}; \mu_{\text{data}}, \sigma_{\text{data}}^2 + \sigma^2(t)\right) \quad (13)$$

The corresponding score function of this noise-perturbed distribution then becomes:

$$\mathbf{s}_t \triangleq \nabla_{\mathbf{x}} \log p_t(\mathbf{x}) = \frac{\mu_{\text{data}} - \mathbf{x}}{\sigma_{\text{data}}^2 + \sigma^2(t)} \quad (14)$$

Now we consider a low-temperature sampling scheme, where the original data distribution is transformed to $p_t'(\mathbf{x}) = \frac{1}{Z} p_0(\mathbf{x})^{\frac{1}{T}}$ through a temperature factor $T$. This results in the variance re-scaled by $T$, i.e.

$$p_t'(\mathbf{x}) = \mathcal{N}\left(\mathbf{x}; \mu_{\text{data}}, T\sigma_{\text{data}}^2 + \sigma^2(t)\right) \quad (15)$$

$$\mathbf{s}_t' = \frac{\mu_{\text{data}} - \mathbf{x}}{T\sigma_{\text{data}}^2 + \sigma^2(t)} \quad (16)$$

$$= \lambda_t \mathbf{s}_t \quad \text{, where } \lambda_t = \frac{\sigma_{\text{data}}^2 + \sigma^2(t)}{T\sigma_{\text{data}}^2 + \sigma^2(t)} \quad (17)$$

Strictly solving the re-weight coefficient $\lambda_t$ is meaningless and would only apply to the simplified Gaussian distribution. Following [28], we assume the perturbed distribution's variance is approximately the maximum variance in the VE-SDE $\sigma_{\text{data}}^2 + \sigma^2(t) \approx \sigma_{\text{max}}^2$. Finally we got a re-weight coefficient:

$$\lambda_t = \frac{\sigma_{\text{max}}^2}{T\sigma_{\text{max}}^2 - (T-1)\sigma^2(t)}$$

## F  Confidence Selection

As we formulate the sidechain packing problem in the view of generative modeling, we can sample multiple conformation from the learned conformation distribution. Here we train an additional *confidence module* to select the most likely sample. The idea of self-predicting sampling quality is also adopted by AlphaFold [32], where pLDDT estimate the quality of protein structure.

Specifically, we utilize a model bearing the same architecture as that used in score prediction. Following the computation of the residue-level representation, we augment the model with an additional MLP head, thereby generating a scalar confidence score for each residue. This confidence score is then trained to correspond with the negative residue-level RMSD. During the inference stage, we sample four conformations for each protein and select the one boasting the highest confidence.

To show how confidence selection improves performance, we randomly select 10 proteins from the test set (each having more than 150 residues) and plot the angle accuracy in accordance with an increasing number of samples (Figure 9). Further, we have measured the correlation between the predicted confidence score and the negative RMSD, yielding **Pearson coefficient 0.664** and **Spearman coefficient 0.798**, respectively. These statistics demonstrate that the predicted confidence score serves as an effective indicator of sample quality, thereby justifying its application in our model.

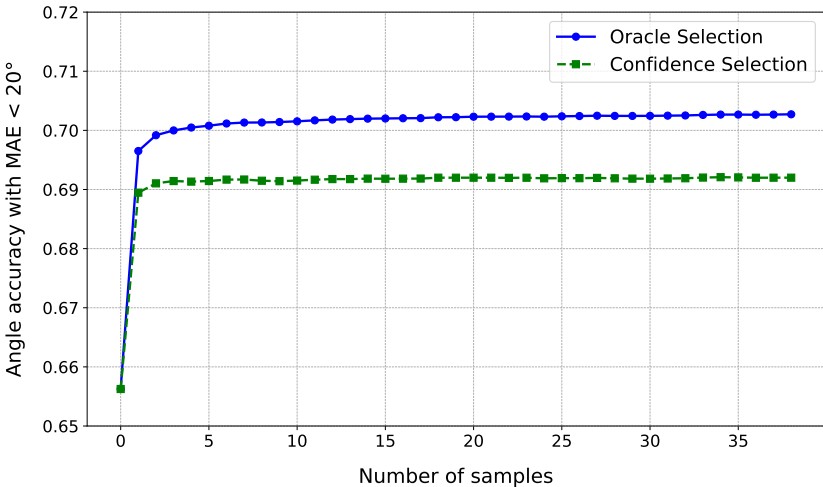

Figure 9: **Percentage of accurate angles as number of samples increasing**. "Oraca Selection" denotes selecting the conformation with lowest RMSD. "Confidence Selection" denotes selecting by trained confidence module.

## G  Inference Time

### G.1  Comparative results of inference time

In this section, we evaluate the inference speed of various methods, categorizing them as GPU-based or CPU-based. All GPU-based methods[1] were evaluated on an NVIDIA RTX A100 40GB GPU, while CPU-based methods were assessed on an AMD EPYC 7513 32-Core Processor @ 2.60 GHz. For algorithms that facilitate batch processing, the batch size was meticulously chosen and fine-tuned to an optimal value, taking into account specific computational requirements.

It's essential to recognize that the inference time for DiffPack is highly influenced by the number of samples in confidence selection and the number of rounds in multi-round sampling. We denote DiffPack-vanilla as the model that samples one conformation without confidence selection. As illustrated in Table 7, DiffPack-vanilla outperforms other GPU-based methods in terms of both speed and performance. Although the CPU-based method FASPR achieves superior speed through judicious

---

[1]The symbol † is used to denote methods that utilize GPU processing.

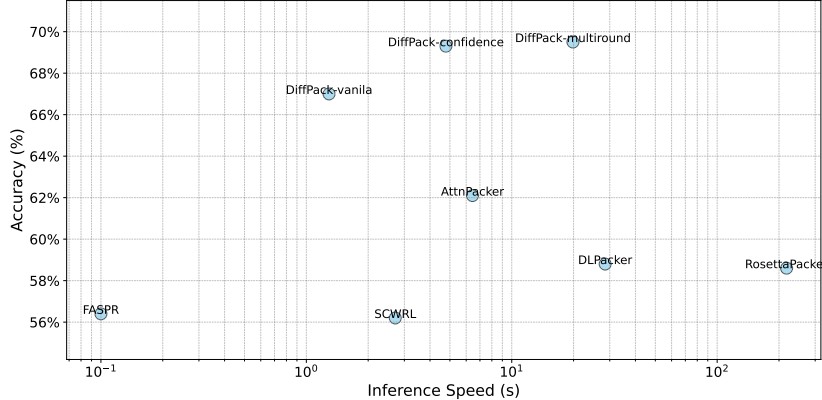

Figure 10: **Comparision of speed and accuracy between different methods.** Inference speed is shown in log-scale

optimization within its tree search algorithm, its restricted angle accuracy limits its applicability in contexts where precision takes precedence over speed especially in sidechain packing.

DiffPack's flexibility allows for integration with various supplementary techniques as shown in Section 3.5, trading off some speed for enhanced performance. For a detailed comparative analysis, we provide a plot (shown in Figure 10) that elucidates the interplay between DiffPack's speed and corresponding performance. Among the variations of DiffPack, we refer to the model amalgamated with confidence selection as DiffPack-confidence, and the one further augmented with multi-round sampling as DiffPack-multiround.

## G.2 Further acceleration on DiffPack

It is noteworthy that DiffPack's speed is primarily constrained by the multi-step nature of the denoising process. Recently, significant research efforts [42, 43] have been directed towards accelerating this process. Pursuing integration with these speed-enhancing methods constitutes an exciting avenue for future research and potential further optimization.

Beyond the algorithm itself, DiffPack's performance is intricately tied to the specific architectural design of GearNet-Edge, which is utilized to learn the score function within the torsion space. This architecture leverages multi-relational message passing within the protein's edge-graph (i.e. line-graph). We have invested some efforts in optimizing the inference speed of GearNet-Edge, based on DGL [62]. Our preliminary findings have yielded promising results, reducing the inference time from **1.29s/protein** to **0.67s/protein** on a single A100 GPU, and to **3.9s/protein** on a single SPR CPU.

| | INFERENCE TIME(S) ↓ | ANGLE ACCURACY ↑ |
|---|---|---|
| SCWRL | 2.71 | 56.2% |
| FASPR | **0.10** | 56.4% |
| RosettaPacker | 217.80 | 58.6% |
| DLPacker[†] | 28.50 | 58.8% |
| AttnPacker[†] | 6.33 | 62.1% |
| DiffPack-vanila[†] | 1.29 | **67.0%** |

Table 7: **Average Inference Time of Different Methods**

# H Experimental Setup

## H.1 Details of Baselines

**SCWRL4.** SCWRL4 [35] is a widely used and well-established method for protein side-chain conformation prediction. It employs a graph-based approach, where side chains are represented as

nodes in a graph, and edges connect pairs of nodes that have potential steric clashes. The algorithm performs a combinatorial search to find the most probable side-chain conformation with minimal steric clashes and optimal energy, using a backbone-dependent rotamer library and a statistical potential energy function derived from known protein structures.

**FASPR.** FASPR (Fast and Accurate Side-chain Prediction using Rotamer libraries) [27] is a prediction method that leverages backbone-dependent rotamer libraries [53] and a custom-built energy function to efficiently predict side-chain conformations. The method involves using Dead-End Elimination (DEE) and tree decomposition to find the set of rotamers that allows the protein to adopt the Global Minimum Energy Conformation (GMEC). FASPR is designed to achieve a balance between computational speed and accuracy, making it suitable for large-scale protein modeling applications.

**RosettaPacker.** RosettaPacker [8] is a component of the Rosetta molecular modeling suite, which is widely recognized for its versatility and accuracy in various applications, including protein structure prediction, protein-protein docking, and protein design. The RosettaPacker algorithm utilizes Monte Carlo simulations combined with a detailed all-atom energy function to sample side-chain conformations and optimize packing interactions. The method is known for its ability to explore a broad conformational space, making it particularly effective for challenging prediction tasks that involve significant side-chain rearrangements.

**DLPacker.** DLPacker [46] is a deep learning-based method for side-chain conformation prediction, which employs convolutional neural networks (CNNs) to predict the atom density map of side-chain conformation from the amino acid sequence and backbone conformation. The method incorporates local and non-local context information from the protein structure to make predictions, and it is trained on a large dataset of high-resolution protein structures. By leveraging the power of deep learning, DLPacker can capture complex sequence-structure relationships, leading to improved prediction accuracy.

**AttnPacker.** AttnPacker [45] is a state-of-the-art method for side-chain conformation prediction that utilizes the attention mechanism, a powerful technique commonly employed in deep learning architectures for tasks involving sequence data. The method incorporates multiple layers of complex triangle attention operations, which enable it to learn long-range dependencies and spatial relationships in protein structures. AttnPacker's architecture allows it to model complex protein conformations with high accuracy; however, its large number of parameters makes it computationally demanding compared to some of the other methods.

Results of baselines are directly taken from the AttnPacker paper [45].

## H.2 Training Details

For torsional diffusion, we utilize the Variance-Exploding SDE (VE-SDE) framework, where $f(\boldsymbol{\chi}, t) = 0$ and $g(t) = \sqrt{\frac{d}{dt}\sigma^2(t)}$. The choice of $\sigma(t)$ follows the exponential decay defined in previous research [57], given by $\sigma(t) = \sigma_{\min}^{1-t}\sigma_{\max}^{t}$ with $\sigma_{\min} = 0.01\pi$, $\sigma_{\max} = \pi$, and $t \in (0, 1)$. During training, we randomly sample a time step $t$ and embed it with sinusoidal embeddings, which is then fused with node features. For inference, we use 10 time steps interpolated between $\sigma_{\min}$ and $\sigma_{\max}$ for generation. At each time step, we perform four rounds of sampling. Subsequently, for each target, we randomly select four different predictions from our model and employ our confidence model to choose the best prediction. To encode the protein structure and denoise torsional angles, we employ a 6-layer GearNet-Edge model with a hidden dimension of 128. For edge message passing, the connections between edges are divided into 8 bins based on the angles between them.

All the models are trained using the Adam optimizer with a learning rate of 1e-4 and a batch size of 32. The training process is performed on 4 A100 GPUs for 400 epochs.

