# OpenReview forum: "DiffPack: A Torsional Diffusion Model for Autoregressive Protein Side-Chain Packing"
_NeurIPS.cc/2023/Conference — NeurIPS 2023 poster_

### Official Review · Reviewer_uGQU · 2023-06-21

**Soundness:** 3 good
**Presentation:** 4 excellent
**Contribution:** 3 good
**Rating:** 6
**Confidence:** 4

**Summary:**

The paper proposes DiffPack, a torsional diffusion model that accurately predicts the conformation of protein side-chains given their backbones. DiffPack learns the joint distribution of side-chain torsional angles by diffusing and denoising on the torsional space. To avoid issues arising from simultaneous perturbation of all four torsional angles, the paper proposes autoregressively generating the four torsional angles and training diffusion models for each torsional angle. The method achieves remakrable improvements in angle accuracy on benchmark datasets and enhances side-chain predictions in the AlphaFold2 model.



**Strengths:**

-The paper is well-written and easy to follow, with a clear description of the proposed DiffPack model. While the idea of using torsional diffusion is not entirely novel, the paper builds on the work of Jing et al. at NeurIPS 2022 on small molecule torsion diffusion and applies it to the problem of protein side-chain packing.

-The paper's technical contributions lie in the development of a torsional diffusion model that considers the restrictions imposed by covalent bond lengths and angles, and the autoregressive generation of torsional angles.

-The paper's results demonstrate the potential of DiffPack in advancing protein structure prediction and design.



**Weaknesses:**

-Is there a metric available for assessing the overall conformational plausibility of the generated structures? This question is relevant because in the context of small molecule conformation generation, deep learning methods have been shown to produce conformations with lower overall RMSD, but there are still significant challenges with conformational plausibility (e.g. generated benzene rings are not necessarily planar). t would be beneficial to have a metric that takes into account both the geometric accuracy and conformational plausibility of the generated structures to ensure that they are reliable for downstream applications.

-It would be helpful to include a comparison of model run times between DiffPack and other methods such as Attenpack and force field-based methods. This would enable us to better assess the potential gaps in downstream applications when making proteomic predictions with these methods.

**Questions:**

Please see the above weaknesses.

**Limitations:**

The authors didn't discuss the limitations and potential negative societal impact of their work.

---

> ### Author Rebuttal · Authors · 2023-08-10
>
> Thanks for your suggestions! Here is our response to your concern.
> ___
> **Q1: Is there a metric available for assessing the overall conformational plausibility of the generated structures?**
>
> This is an interesting question! Assessing the overall conformational plausibility of generated structures, particularly in complex biological systems like proteins, is a multifaceted task. Researchers usually employ two kinds of  factors to evaluate the plausibility of a conformation.
>
> 1. **Steric Clash Assessment**: Steric clash is the most direct way for assessing the plausibility of generated conformation. Conformation which violates chemical constraints will usually impose unnatural overlap between atoms in 3D space.
>
> 2. **Physical Energy Assessment**: The total potential energy of a conformation can also be used as a proxy for its plausibility. Conformations that are at or near a local minimum of the energy landscape are typically more plausible.
>
> We choose steric clash assessment to evaluate the plausibility of a conformation in our work. This measurement does not rely on manually selected energy functions, making it a more straightforward choice. Specifically, the distance threshold between different atoms are first predefined according to van der Waals radii, and further adjusted to account for different chemical interaction(e.g. H-bond and disulfide bridges). We classify the conformation as a steric clash if the pair distance between atoms violates this threshold. Details can be found in Appendix B.
>
> Furthermore, it's essential to highlight that our proposed method, which operates in torsion space, has distinct advantages over models operating in Cartesian space, such as AttnPacker. By treating the functional group as a whole in torsion space, intra-group implausibilities (e.g. non-planar generated benzene rings) are unlikely, leaving only inter-group implausibilities. This inherent characteristic of our approach emphasizes its superiority in ensuring conformational plausibility.
> ___
> **Q2: It would be helpful to include a comparison of model run times between DiffPack and other methods such as Attenpack and force field-based methods.**
>
> Thanks for your valuable suggestions! We recognize the importance of comparing inference times between our method and other established approaches, and we've included a detailed comparison in global rebuttal block’s attachment. A summary of the experiment results is presented below:
>
> | Methods | Inference Time (s) ↓ | Angle Accuracy ↑ |
> |---|:---:|---|
> | SCWRL | 2.71 | 56.2% |
> | FASPR | **0.10** | 56.4% |
> | RosettaPacker | 217.80 | 58.6% |
> | DLPacker* | 28.50 | 58.8% |
> | AttnPacker* | 6.33 | 62.1% |
> | DiffPack_vanila* | 1.29 | **67.0%** |
>
> As evident from the table, DiffPack_vanila outperforms other GPU-based deep learning methods (denoted by *) in terms of speed while achieving the highest angle accuracy. Although the CPU-based method FASPR is quicker due to efficient optimizations in the tree search process, its prediction accuracy lags behind. In the sidechain packing task, where prediction accuracy is prioritized over inference time, DiffPack's performance becomes particularly significant.
>
> This comparison highlights DiffPack’'s balance between computational efficiency and prediction accuracy, showcasing its viability and strength as a choice for this specific application. The comprehensive analysis can be found in the global rebuttal block for further insights. We will also include it in the revised version of the paper.

---

> > ### Comment · Reviewer_uGQU · 2023-08-14
> > **Thanks for the rebuttal**
> >
> > Thank you for the response. It well addressed my concerns.

---

> > > ### Author Response · Authors · 2023-08-18
> > > **Thanks**
> > >
> > > We are very glad that our responses is able to address your concerns. Thank you again for your time and patience!

---

### Official Review · Reviewer_KyoN · 2023-06-22

**Soundness:** 3 good
**Presentation:** 3 good
**Contribution:** 2 fair
**Rating:** 6
**Confidence:** 4

**Summary:**

The paper proposes DiffPack a torsional diffusion model to learn side chain placements. In particular, the authors presents a few modification to vanilla torsional diffusion models that improve the empirical results obtaining strong empirical performance.

**Strengths:**

The authors propose a number of modifications to the vanilla application of torsional diffusion to side chain predictions and these have enables them to obtain state-of-the-art performance on this important task.

**Weaknesses:**

Although the empirical results are very good, not many parts of the method are significantly novel. Moreover, among the few modifications that the authors made the presentation of them with respect to the rest of the field and the justification should be improved:

1. Annealed temperature sampling: the low temperature sampling procedure derived from the assumption of the Gaussian distribution seems to be very similar to the one presented in Ingraham et a. (2022) [20]. The authors should include the reference of this technique in the relevant section and clarify the difference between the methods. Moreover, I believe the statement “ideally the sampling process converges to the global optimum when T→0” is wrong or at least misleading (does not specify what the ideal situation is).

2. Multi-round sampling: this is related to the mixing Langevin steps that multiple papers have proposed before (e.g. Ingraham et a. [20]). However, unlike in those approaches, the authors do not limit themselves to a Langevin step that preserves the expected distribution but instead keep the “ODE-term” making therefore very unclear what the process is “theoretically” doing in this process. The authors should provide further discussion of these challenges.

3. Autoregressive diffusion: some of the comparisons (excluding the inference performance) used to motivate autoregressive diffusion versus vanilla are misleading:
 - Figure 7: presents the number of steric clashes during the diffusion process not of the generated structures. This is misleading because these steric clashes may not matter or even help the model to detect them to avoid them. As the autoregressive method (that removes following atoms) artificially removes these steric clashes and this may prevent it to reason about them when generating the distributions. The number of steric clashes should be compared based in the resulting generated structures.
 - Figure 5 compares the loss values of different methods however these are not comparable. E.g. one of the reason for the loss of X1 is lower than X4 might just be fact that the entropy of the distributions of X1 (which is correlated to the lower bound of the score matching loss) is lower than that of X4. Similarly this makes the comparisons of the losses of the different methods not comparable (e.g. the entropy of the conditional distribution obtained by the autoregressive is lower than that of the unconditional obtained by the joint loss).

**Questions:**

The statement “We focus exclusively on protein side-chain prediction under the assumption of a fixed and highly accurate backbone” seems to contradict with the AlphaFold2 results presented in the main text.

“We extend DiffPack to accommodate non-native backbones generated from AlphaFold2”. How is this extension done? What is the model trained on?

Section B.1 what is the distance threshold?

**Limitations:**

In some of the tables the (e.g. Table 3) the bolding of the best number is wrong and biased to the proposed method.

I believe that the discussion on the number of parameters is misleading as this is not a very useful measure of computational feasibility. Instead the authors should compare the methods by runtime (and potentially memory cost).

---

> ### Author Rebuttal · Authors · 2023-08-10
>
> Thank you for your insightful questions. We'd like to first clarify that our work's main focus is on formulating a new approach to the sidechain packing task, rather than proposing general improvements to the diffusion process. Below are our specific responses to your queries.
> ___
> **Q1: Justification about annealed temperature sampling**
>
> The annealed temperature sampling used in our work indeed has similarities to Ingraham et al. (2022) [28]. While the core concept is borrowed from this previous work, we specifically derived the VE-SDE version of the annealed weight under the same assumption. Although initially referenced in Appendix E, we understand the explanation may have been unclear. Therefore, we'll enhance the relevant section in the main text for better clarity, in line with your valuable suggestion.
>
> Regarding the statement, “ideally the sampling process converges to the global optimum when T→0”, we understand how this can be misconstrued, and we appreciate your attention to the detail.  What we intended to convey is that theoretically, the distribution $p_T(\mathbf{x})$ would collapse to a Dirac delta function at the global optimum as **temperature $T$ approaches zero**. However, this is only an approximation in practical applications. We will rewrite this section to eliminate any ambiguity.
>
> Thanks again for your valuable suggestions.
> ___
> **Q2: Justification about multi-round sampling**
>
> Thanks for pointing this out! Upon a closer review of the literature[1], we found that the multi-round sampling is indeed closely connected to langevin dynamics.
>
> Since we utilize VE-SDE which neglect the drift term, the reverse VE-SDE differs from mixing langevin dynamic[1] by only a coefficient $\sqrt{2}$. This coefficient may be interpreted as a temperature scaling factor, and we hypothesize that it contributes to the improvement in the original multi-round sampling.
>
> But as you mentioned, the proposed multi-round sampling is indeed a special case of the general mixing langevin dynamics. In light of this, we further conducted a new experiment with the more general mixing langevin dynamics, observing improvements in key metrics, such as an increase in Angle Accuracy from 69.5% to 70.1% in CASP13. These new findings will be incorporated into the revised version of our paper.
>
> Your feedback has provided invaluable guidance, enabling us to refine our method and elucidate the underlying mechanisms. We are sincerely grateful for your contributions to the advancement of this work. Thank you again!
>
> [1] Song, Yang, et al. "Score-based generative modeling through stochastic differential equations." ICLR 2020.
> ___
> **Q3:  The number of steric clashes during the diffusion process is misleading.**
>
> Thank you for raising this concern!  We appreciate your insights and would like to clarify that Figure 7 primarily focuses on the training phase rather than the inference process. While it’s true that the steric clashes might be managed by a well-trained model during inference, it is important to emphasize that they pose significant challenges during the training phase. Autoregressive diffusion is designed to alleviate these training difficulties, which might not be immediately apparent during the inference period.
>
> Your suggestion to compare steric clashes based on the resulting generated structures is great. We evaluated the clash number within 90% distance threshold, finding that joint diffusion did not reduce more clash pairs (Joint: 6.9 vs Autoregressive: 6.0). Furthermore, as shown in Table 4, the substantial improvements in practical applications achieved by autoregressive diffusion over joint diffusion demonstrate that autoregressive diffusion is a better choice in practical application.
> ___
> **Q4: Loss values of different methods in Fig. 5 are not comparable.**
>
> Thank you for bringing this to our attention! We acknowledge that the loss values comparison in Figure 5 might indeed be misleading due to differing entropies as lower bounds across the methods, rendering the losses not directly comparable. We will certainly revise the figure in line with your valuable suggestions.
>
> It's also worth emphasizing that the average training loss of autoregressive diffusion is bounded by the entropy of distribution $p\left(\boldsymbol{\chi}_1, \boldsymbol{\chi}_2, \boldsymbol{\chi}_3, \boldsymbol{\chi}_4\right)$, as is the loss of joint diffusion. While the original comparison might have been imprecise, we believe that a rough comparison can still shed light on the relative optimization or training efficiency of autoregressive modeling versus joint diffusion. Coupled with the experimental results shown in Table 4, these findings affirm the efficacy of autoregressive modeling.
> ___
> **Q5: The statement “We focus exclusively on ….” seems to contradict with the AlphaFold2 results presented.**
>
> Thank you for raising the concern! The additional experiments with AlphaFold2 aim to showcase potential usages of our method on non-native backbones, which does not change the main focus of the paper. We will tune our statement in the revised version.
> ___
> **Q6: How to extend DiffPack on non-native backbones?**
>
> We simply adapt the DiffPack model trained on native backbones to AF-predicted backbones without any retraining. The results actually showcase the generalization ability of the proposed method. We will clarify this point in the revised version.
> ___
> **Q7: In Sec. B, what is the distance threshold?**
>
> Following Matthew et al.[45], the distance threshold between different types of atoms is initially defined by van der Waals radii and further adjusted to account for factors such as H-bond and disulfide bridges. We will add more details to related sections.
> ___
> **Q8: Issues about computational feasibility**
>
> Thanks for your question! We’ve conducted an experiment for the inference speed between different methods. Details can be found in attached materials and global rebuttal block.

---

> > ### Comment · Reviewer_KyoN · 2023-08-15
> > **Response to rebuttal**
> >
> > Thank you for the response, I am glad my comments were helpful and I appreciate the effort in clarifying the relation of the method to previous work. I have raised my score to 6.

---

> > > ### Author Response · Authors · 2023-08-18
> > > **Thanks**
> > >
> > > We're glad that our replies have been able to address your concerns, and we sincerely appreciate your efforts to enhance the quality of the work. Thank you once again!

---

### Official Review · Reviewer_Npgs · 2023-07-04

**Soundness:** 3 good
**Presentation:** 3 good
**Contribution:** 2 fair
**Rating:** 5
**Confidence:** 3

**Summary:**

This paper focuses on the task of sidechain packing in proteins, where one wishes to predict the positions of the side-chain atoms given the positions of the backbone atoms. To this end, the main contribution of the paper is DiffPack, an extension of recently proposed torsional diffusion models to the side-chain packing task. This is achieved by combining three aspects:

i) autoregressive sampling of the four side-chain torsion angles, with a separate diffusion model trained for each, conditioned on the previous torsion angles

ii) multi-round and annealed temperature sampling to improve quality of generated samples

iii) Modifications to the transition kernel formulation on the torus for residues where the torsion angles have a periodicity of $\pi$.

Experimental performance compared to various baselines for side-chain sampling showcases the improved performance offered by DiffPack.

**Strengths:**

1. The paper is very well written, clear and easy to understand. The technical choices made throughout the paper are sound and well motivated, and the application area considered is well suited for the torsional diffusion framework.

2. The authors compare their method to a variety of baselines for side-chain packing and showcase improved experimental performance of their method.

**Weaknesses:**

1. The technical contributions are largely incremental - the formulations regarding torsional diffusion have already been well explored in previous recent papers.

2. Some questions / clarifications regarding the experimental evaluation:

    * It is unclear to me the benefits offered by using auto-regressive models for each torsion angle as opposed to joint diffusion. From Table 1, 2 and 4, one can see that, for about 4$\textdegree$ variation in MAE of $\chi_1$, the RMSD drops by 0.1. From Table 4, the $\chi_1$ MAE between the autoregressive model and the joint diffusion model is about 2$\textdegree$, giving a much smaller corresponding drop in RMSD. Using auto-regressive models definitely seems to accelerate training / convergence as noted in Fig 5. However, the errors in discretization during sampling, idealization of bond lengths and angles when reconstructing coordinates could end up reducing the improved training performance of the autoregressive models, and eventually offer similar values in RMSD.

    * Could the authors add a table regarding the run times associated with the evaluations of the different deep learning baselines? How many samples are generated for each protein before selection with the confidence model?

3. An anonymous link for the code submission is not provided, making it harder to verify reproducibility.

**Questions:**

See the above section for questions.

Edit: I have read the authors rebuttal, and post the discussion phase, still maintain my assessment about the paper.

**Limitations:**

The authors have addressed limitations associated with their work

---

> ### Author Rebuttal · Authors · 2023-08-10
>
> Thanks for your suggestions! Here is our response to your concern.
> ___
> **Q1: Potential Concern in Incremental Technical Contribution**
>
> Thanks for your review! Formulation of diffusion models on Riemannian manifold(e.g. torsion space) was first proposed by Valentin De Bortoli[1] and further applied in various domains such as molecule conformation generation[2], antibody optimization[3], molecule docking[4] and protein backbone design[5]. However, to our knowledge, **DiffPack represents the first work of Riemannian diffusion model in the domain of protein side chain packing**, which significantly improves the prediction accuracy, as demonstrated in our experiments.
>
> 1. **Compared with other work where the Riemannian diffusion model is applied, protein side-chain packing presents a much higher degree of freedom.** Earlier studies have focused on either small molecules [2][4] (averaging ~50 atoms) or residue-level proteins [5] (averaging ~200 residues). In our case, we had to model an average of 3000 atoms. Even though we constrained the degree of freedom by choosing the torsion space as the model output, this complexity still far exceeds previous scenarios.
>
>    Coupled with the issues discussed in Section 3.3 (Cumulative Coordinate Displacement and Excessive Steric Clash), this complexity posed a significant challenge to directly train and sample from a diffusion model in this space. As shown in Figure 5, directly optimizing a joint diffusion model suffers from the underfitting problem. To address this problem specific challenge, autoregressive modeling is introduced to factorize the joint distribution into a product of conditional distribution. This significantly alleviated the under-fitting problem.
>
> 2. **Compared with previous work in protein side-chain packing, DiffPack is the first method modeling the side-chain problem in a generative manner.** Previous regression based methods (AttnPacker, DLPacker) tend to predict the “mean conformation”, while our proposed method could capture the entire distribution of side-chain conformation. Experiments have validated the effectiveness of our generative modeling.
>
> [1]. De Bortoli, Valentin, et al. "Riemannian score-based generative modelling." Advances in Neural Information Processing Systems 35 (2022): 2406-2422.
>
> [2]. Jing, Bowen, et al. "Torsional diffusion for molecular conformer generation." Advances in Neural Information Processing Systems 35 (2022): 24240-24253.
>
> [3]. Luo, Shitong, et al. "Antigen-specific antibody design and optimization with diffusion-based generative models for protein structures." Advances in Neural Information Processing Systems 35 (2022): 9754-9767.
>
> [4]. Corso, Gabriele, et al. "Diffdock: Diffusion steps, twists, and turns for molecular docking." International Conference on Learning Representations (2023).
>
> [5]. Wu, Kevin Eric, et al. "Protein structure generation via folding diffusion." (2022).
> ___
> **Q2: Issues regarding the benefit of autoregressive modeling**
>
> We appreciate your insights and would like to elaborate on the advantages of autoregressive modeling in our context. Protein sidechain packing indeed poses a formidable challenge due to its vast degree of freedom. When training a model on the joint distribution directly, the computational complexity can grow exponentially with the number of variables due to the "curse of dimensionality". Besides, the additional challenges(Cumulative Coordinate Displacement and Excessive Steric Clash) also impose negative influence for directly modeling the joint diffusion process. All of the above will finally lead to an underfitting problem. So in our work, autoregressive is introduced to alleviate this problem. By decomposing the joint distribution into conditional distributions, we simplified the denoising process, enabling us to capture the potential spatial dependencies between $\chi_k$ and $\chi_{i<k}$.
>
> Regarding ablation study performance, we respectfully disagree with the reviewer about the improvement achieved by our method. It can be clearly observed that the MAEs of all four angles significantly drop after replacing joint diffusion with autoregressive diffusion on CASP14. Additionally, we provide Atom RMSD results in Table A, illustrating a reduction of approximately 0.12 in RMSD, which is already significant in sidechain packing.
> Table A: Ablation Study on CASP14.
> |#Method|$\chi_1$ MAE|$\chi_2$ MAE|$\chi_3$ MAE|$\chi_4$ MAE|Atom RMSD|
> |:----:|:----:|:----:|:----:|:----:|:----:|
> |**DiffPack**|**21.91**|**25.54**|**44.27**|**55.03**|**0.770**|
> |w/ joint diffusion|26.80|34.51|52.77|63.41|0.893|
>
> Finally, it's worth noting that autoregressive models aren't necessarily "better" or "worse" than other types of models. They are simply better suited to certain types of problems
> ___
> **Q3: Issues regarding running times**
>
> Thanks for your valuable suggestions! We have included the running time information in Table 2 of the attachment. In short, DiffPack is the best model for balancing speed and accuracy. You can find more details in the global rebuttal section and attachment.
>
> | Methods | Inference Time (s) ↓ | Angle Accuracy ↑ |
> |---|:---:|---|
> | SCWRL | 2.71 | 56.2% |
> | FASPR | **0.10** | 56.4% |
> | RosettaPacker | 217.80 | 58.6% |
> | DLPacker* | 28.50 | 58.8% |
> | AttnPacker* | 6.33 | 62.1% |
> | DiffPack_vanila* | 1.29 | **67.0%** |
> ___
> **Q4: How many samples are generated for each protein before selection with the confidence model?**
> Thanks for your questions! We sample 4 conformations for each protein. The value is chosen to balance the speed and accuracy as shown in the attached Figure 1. Additional details of inference process can be found in Appendix F.2
> ___
> **Q5: Issue about code and reproducibility**
> We have attached the anonymous code link(https://anonymous.4open.science/r/DiffPack-DED9/). Thank you for raising the concern!

---

> > ### Comment · Reviewer_Npgs · 2023-08-16
> > **Response to Rebuttal**
> >
> > I thank the authors for their efforts with the rebuttal process. Below follows a point-by-point response to the rebuttal:
> >
> > **Q1**
> > Thank you the comments, the increase in scale is indeed a valid point which I missed to include before, but some concerns proposed to deal with the scale still remain (more in Q2).
> >
> > **Q2**
> > Thank you for the clarification! I understand the issues associated with sidechain packing as you outlined and the utility of autoregressive models in this situation, but if they were to indeed mitigate the issues to the extent as expected, the peformance should be considerably better. Could you provide some references as to ranges of RMSD values in sidechain packing that qualify as poor vs satisfactory?. In the protein structure prediction tasks, <1A RMSD is already considered near native for instance.
> >
> > For a task utilizing diffusion models, 0.12A RMSD could just be a byproduct of the noise scale at the final step.
> >
> > **Q3, Q4 and Q5**
> > Thank you, these experiments definitely shed more light on the utility of the DiffPack as a fast & accurate sampler.
> >
> > Time permitting question this, but have the authors carried out any quantitative analysis of the steric effects in the predictions from models with and without autoregressive diffusion?

---

> > > ### Author Response · Authors · 2023-08-18
> > > **Thanks for your comment!**
> > >
> > > Thank you for your thoughtful comments and questions. We appreciate the opportunity to provide further clarifications and context for our work.
> > >
> > > In protein sidechain packing,  Angle Mean Absolute Error (Angle MAE) and Atom Root Mean Square Deviation (Atom RMSD) are commonly used metric. As cited in references [1][2][3], a sidechain is considered to be accurately predicted when the error is less than or equal to a predefined threshold, which is typically 20° or 40°. In our work with DiffPack, the model achieved an angle accuracy of **57.5%** using a 20° threshold, compared to **49.9%** for DiffPack without autoregressive diffusion.
> > >
> > > To further clarify your concern about RMSD value, we would like to emphasize that, in the context of protein sidechain packing, the predicted atom positions are confined to the region of the residue. As a result, the RMSD values are expected to be significantly lower than those observed in protein protein backbone-related studies. To put it into perspective, even traditional sidechain packing methods have been known to achieve around 1Å RMSD, which may seem unexpectedly low when compared to backbone prediction tasks.
> > >
> > >
> > > [1]. Dunbrack Jr, Roland L., and Martin Karplus. "Backbone-dependent rotamer library for proteins application to side-chain prediction." Journal of molecular biology 230.2 (1993): 543-574.
> > >
> > > [2]. Colbes, José, et al. "Protein side-chain packing problem: is there still room for improvement?." Briefings in bioinformatics 18.6 (2017): 1033-1043.
> > >
> > > [3]. Jumper, John, et al. "Highly accurate protein structure prediction with AlphaFold." Nature 596.7873 (2021): 583-589.
> > >
> > >
> > > >**Time permitting question this, but have the authors carried out any quantitative analysis of the steric effects in the predictions from models with and without autoregressive diffusion?**
> > >
> > > Thanks for pointing out this! We have conducted an experiment to evaluate the steric clash in structures generated by models with and without autoregressive diffusion. Specifically, when assessing the number of clashes within a 90% distance threshold, we observed that both of them have achieved relatively low number of clash pairs (**6.9** for the model without autoregressive diffusion, and **6.0** for the model with autoregressive diffusion). We will include these findings in our revised manuscript following your valuable suggestions.

---

### Official Review · Reviewer_cTo4 · 2023-07-05

**Soundness:** 4 excellent
**Presentation:** 4 excellent
**Contribution:** 3 good
**Rating:** 7
**Confidence:** 4

**Summary:**

The protein side-chain packing problem consists of predicting the positions of atoms in amino-acid side chains given the backbone structure and residue identities. The paper proposes to do this using a diffusion model that accounts for physical constraints and models side chain structures as joint distributions over torsional angles. Furthermore, they find that generating the torsional angles autoregressively improves the generation quality. DiffPack outperforms existing methods with fewer model parameters and can enhance side chain predictions from AlphaFold2.

**Strengths:**

The paper provides a unique solution to the important and well-studied protein side-chain packing problem. The writing is generally clear, and the work is well-contextualized in the literature. The autoregressive diffusion framework is intuitively effective, and the formulation seems correct. This is backed up by strong empirical results when comparing to previous work and in the ablations. Providing confidence scores, reducing the number of parameters compared to previous models, and being able to refine AlphaFold2 predictions will make the method very useful for downstream practitioners.

**Weaknesses:**

In general, this is a strong submission with few major weaknesses.

### Major

- Clarity: it's a bit unclear to me how the model moves between atomic coordinates in GearNet and predicting the scores and confidences on each angle. A few more lines of text or a subfigure could be very helpful here. Likewise, it would be nice to have some more details about model size, training hyperparameters, and training hardware.
- Soundness: The paper talks about confidence scores and shows that they improve generations. It would be better to also have a table or figure showing how well the confidence scores are calibrated.
- Soundness: are the baselines also given the chance to generate multiple conformations and then to have a confidence model pick the best one? If not, the comparisons are not quite one-to-one.
- Significance: While I appreciate the case studies shown in 5.5, the paper would be stronger with more context here. What downstream biological or engineering applications, if any, can DiffPack do that are not accessible with existing methods?

### Minor

- There are a few minor points on clarity: The standard in the field seems to be DDPM (denoising diffusion probabilistic models) instead of DPM, as used in line 252. In Figure 4, the legend says blue and yellow, but I see blue and red in the figure. In Figure 5, the colors for the four autoregressive curves are very difficult to tell apart -- it might help to also vary the linestyle.

**Questions:**

- Is there any intuition for choosing the variance exploding SDE instead of variance preserving?
- How well-calibrated are the confidence predictions?
- What hardware was used to train the model? How long was it trained for?
- What downstream biological or engineering applications, if any, can DiffPack do that are not accessible with existing methods?
- How exactly does the model move between atomic coordinates and score / confidence predictions?
- When generating, how many different conformations are being generated and passed to the confidence model?
- What work remains to be done in side-chain packing? Does DiffDock completely solve the problem?
- How hard would it be to train an order-agnostic autoregressive diffusion here? Would we expect that to produce better samples than a model with a fixed decoding order, as shown here?

**Limitations:**

- The authors should address where DiffDock still does not solve the side chain packing problem, either in general or in specific cases.

---

> ### Author Rebuttal · Authors · 2023-08-10
>
> Thanks for your time and review! We think that the majority of your detailed concerns about our method are covered in the Appendix. We'll revise the order as you've recommended to prevent any confusion. Here's our response to your concerns.
> ___
> **Q1: Clarity about GearNet Score Prediction**
>
> For atom representations, we build a graph combining bonds and 3D coordinates. GearNet-Edge is used on this graph. Residue representations are obtained by averaging atom embeddings. These feed into an MLP for score function and confidence score output. We've addressed this in Sec. 3.4 and will provide more details.
> ___
> **Q2: Calibration of Confidence Scores**
>
> We've done extra experiments to assess the confidence score calibration in the global rebuttal, as per your suggestions.
> ___
> **Q3: Are the baselines also given the chance to generate multiple conformations？ If not, is the comparison fair?**
>
> Thanks for noting this! In the benchmark, traditional methods like SCRRL4, FASPR, and RosettaPacker produce multiple conformations, choosing the best based on an energy function. In contrast, deep learning methods except DiffPack are end-to-end regression models, yielding one pose at a time.
>
> It’s worth noting that DiffPack is the first deep learning method that solves the problem with generative modeling. That’s also one reason that makes our method more effective than others – we capture **the whole conformation distribution** instead of “**the mean conformation**”. As such, predicting side-chain conformation essentially involves sampling from high probability regions of the distribution, while confidence selection helps in identifying these regions.
>
> In contrast, regression-based models such as AttnPacker and DLPacker directly predict the final pose as a result of their regression-based intrinsic design. So there is no need for these models to generate multiple conformations
>
> To further address your concern about potentially unfair comparisons between regression-based and generative models, we've conducted two additional experiments for a more direct comparison shown in attachment Tab. 1:
>
> 1. **Limiting DiffPack’s generation samples to 1**: We constrain DiffPack to generate only one sample, excluding confidence score selection. From the experiment results below, we can see that our models still outperforms other methods by large margin. This is due to the fact that sampling from a probability distribution is typically biased towards regions of higher probability density, without any additional selection module.
> 2. **Allowing other deep learning based models to generate multiple samples**: Despite the intrinsic design of regression-based models not supporting multiple conformation generation directly, varied seed and initialization can lead to different generation results. In this experiment, we enabled these models to generate multiple samples with oracle selection. Even when selecting the best predicted pose based on the ground truth RMSD, the overall results remained largely unchanged, with DiffPack outperforming the competition.
>
> We hope this detailed explanation and the additional experiments satisfactorily address your concern!
> ___
> **Q4: What downstream biological or engineering applications, if any, can DiffPack do that are not accessible with existing methods?**
>
> Essentially, DiffPack does not directly solve a new problem, but instead enhances sidechain packing with significant performance improvement. This allows for applications in areas requiring accurate modeling of sidechain conformation, such as protein-protein docking and protein mutation prediction.  Further details on potential applications are available in App. A.1.
> ___
> **Q5: What's the intuition behind choosing the variance exploding SDE instead of variance preserving?**
>
> The choice of a variance exploding stochastic differential equation (VE-SDE) over variance preserving (VP-SDE) is primarily due to the nature of non-Euclidean torsion space $\mathbb{T}$, where defining an origin isn't straightforward. In traditional Euclidean spaces, the drift term $f(x,t) = -1/2x$ of VP-SDE can target the origin, but in $\mathbb{T}$, this isn't easily feasible. With VE-SDE, the drift term is neglected, making it a favored option in these contexts. While other options exist, VE-SDE is one of the simplest and most effective solutions available.
> ___
> **Q6: What hardware was used to train the model? How long was it trained for?**
>
> The training of our model was conducted on 4xA100 GPUs. The training spanned over a total of 400 epochs, with the entire process taking approximately 4 days for each model. Specific training details are included in App. F.2.
> ___
> **Q7: How many different conformations are being generated and passed to the confidence model?**
>
> Our approach samples four different conformations from the diffusion models during sampling, which are subsequently evaluated by the confidence model. More details are provided in App. F.2.
> ___
> **Q8: What work remains to be done in side-chain packing? Does DiffPack completely solve the problem?**
>
> Regarding side-chain packing, our model is accurate for various applications, but there's room for improving side-chain conformation prediction. For instance, enhancing results on non-native AlphaFold2 backbones is crucial, especially for generating complete atomic conformations in de novo designed sequences.
> ___
> **Q9: Is it promising to train an order-agnostic autoregressive diffusion here?**
>
> While an order-agnostic autoregressive diffusion model has been proposed in previous work[1], we believe it is not suitable for the domain of sidechain packing. As elucidated in Sec. 3.3, the challenge of cumulative coordinate displacement can be mitigated only by fixing the order from $\chi_1$ to $\chi_4$. Without this specific ordering, the underlying issues would persist and likely hinder the training process of the model.
>
> [1]. Hoogeboom et al. "Autoregressive diffusion models." arXiv, 2021.

---

> > ### Comment · Reviewer_cTo4 · 2023-08-10
> > **Response to rebuttal**
> >
> > Thank you for the thorough rebuttal. I am especially pleased with the analysis of the confidence scores and multiple chances for baseline methods. Pending discussion with the other reviewers, I am inclined to raise my score to 7.

---

> > > ### Comment · Reviewer_cTo4 · 2023-08-17
> > > **Score changed**
> > >
> > > I have raised my score from 6 to 7.

---

> > > > ### Author Response · Authors · 2023-08-18
> > > > **Thanks**
> > > >
> > > > We appreciate the opportunity to address your concerns and are grateful for your valuable input, which has greatly enhanced the quality of our work. Thank you again for your effort!

---

### Author Rebuttal · Authors · 2023-08-10

We would like to extend our sincere gratitude for your time and thorough review of our paper. Your insightful suggestions and comments have provided us with valuable perspectives, enabling us to enhance the quality of our work. As there is some common issues raised, we choose to respond to them in the global block.
___
**Common Issue 1: Reproducibility of DiffPack**

Acknowledging the necessity for reproducibility, we have attached the anonymous code (https://anonymous.4open.science/r/DiffPack-DED9/) at this stage. We believe this will address concerns and foster reproducibility within this field.
___
**Common Issue 2: Calibration of Confidence Scores in Confidence Selection**

Several reviewers have inquired about the specifics of confidence selection. To clarify, we conducted additional experiments to demonstrate how confidence selection elevates sampling quality and to assess the calibration of the predicted confidence score.

Specifically, we selected 10 proteins randomly from the test set (each with over 150 residues) and plotted angle accuracy against an increasing number of samples. As depicted in Figure 1, the sampled conformation quality consistently improves with an increased number of samples, notably when the number of samples is <= 5, substantiating the efficacy of our proposed confidence selection.

To further demonstrate how well-calibrated is the predicted confidence score, we have measured the correlation between the predicted confidence score and the negative RMSD, yielding **Pearson coefficient 0.664** and **Spearman coefficient 0.798**, respectively. We plan to present these results in the revised version of paper following your valuable suggestions.
___
**Common Issue 3: Inference Speed and Computation Feasibility in DiffPack**

The computational feasibility of DiffPack has garnered attention from many reviewers.We  performed supplementary experiments to benchmark the inference speed of different methods. Methods are categorized as GPU-based or CPU-based. All GPU-based methods(DLPacker, AttnPacker, DiffPack) were evaluated on an NVIDIA RTX A100 40GB GPU, while CPU-based methods were assessed on an AMD EPYC 7513 32-Core Processor @ 2.60 GHz. For algorithms that facilitate batch processing, the batch size was meticulously chosen and fine-tuned to an optimal value, taking into account specific computational requirements.

It's essential to recognize that the inference time for DiffPack is highly influenced by the number of samples in confidence selection and the number of rounds in multi-round sampling. We denote DiffPack-vanilla as the model that samples one conformation without confidence selection. As illustrated in attached Table 2, DiffPack-vanilla outperforms other GPU-based methods in terms of both speed and performance. Although the CPU-based method FASPR achieves superior speed through judicious optimization within its tree search algorithm, its restricted angle accuracy limits its applicability in contexts where precision takes precedence over speed especially in sidechain packing.

DiffPack's flexibility allows for integration with various supplementary techniques as shown in Section 3.5, trading off some speed for enhanced performance. For a detailed comparative analysis, we provide Figure 2 that elucidates the interplay between DiffPack's speed and corresponding performance. Among the variations of DiffPack, we refer to the model amalgamated with confidence selection as DiffPack-confidence, and the one further augmented with multi-round sampling as DiffPack-multiround.

It is noteworthy that DiffPack's speed is primarily constrained by the multi-step nature of the denoising process. Significant research efforts[1][2] have been directed towards accelerating this process. Pursuing integration with these speed-enhancing methods constitutes an exciting avenue for future research and potential further optimization.

[1]. Song, Jiaming, Chenlin Meng, and Stefano Ermon. "Denoising diffusion implicit models." arXiv preprint arXiv:2010.02502 (2020).
[2]. Lu, Cheng, et al. "Dpm-solver++: Fast solver for guided sampling of diffusion probabilistic models." arXiv preprint arXiv:2211.01095 (2022).

___

Once again, we wish to express our profound gratitude for the meticulous review and constructive feedback. Your guidance has been instrumental in refining our work, and we welcome any further questions or comments you may have.

---

### Decision · Program_Chairs · 2023-09-21

**Decision:**

Accept (poster)

**Comment:**

This is a well-executed paper. It studies the known protein design problem of sidechain packing and brings forth a sensical approach based on recent advances in denoising diffusion, namely torsional diffusion, autoregressive generation, and temperature annealing. The experimental results convincingly demonstrate that the proposed approach can place side chains more precisely than previous baselines. A main criticism one could place is that the methodological components that make this work are mostly known and this paper's contribution lies mostly in combining them in light of the problem at hand. Nevertheless, all reviewers agree that the positives outweigh the negatives, urging for acceptance.